# Distinctive roles of translesion polymerases DinB1 and DnaE2 in diversification of the mycobacterial genome through substitution and frameshift mutagenesis

Pierre Dupuy [1], Shreya Ghosh [2], Oyindamola Adefisayo[1,3], John Buglino[1], Stewart Shuman[2] & Michael S. Glickman [1,3] ✉

Antibiotic resistance of *Mycobacterium tuberculosis* is exclusively a consequence of chromosomal mutations. Translesion synthesis (TLS) is a widely conserved mechanism of DNA damage tolerance and mutagenesis, executed by translesion polymerases such as DinBs. In mycobacteria, DnaE2 is the only known agent of TLS and the role of DinB polymerases is unknown. Here we demonstrate that, when overexpressed, DinB1 promotes missense mutations conferring resistance to rifampicin, with a mutational signature distinct from that of DnaE2, and abets insertion and deletion frameshift mutagenesis in homo-oligonucleotide runs. DinB1 is the primary mediator of spontaneous −1 frameshift mutations in homo-oligonucleotide runs whereas DnaE2 and DinBs are redundant in DNA damage-induced −1 frameshift mutagenesis. These results highlight DinB1 and DnaE2 as drivers of mycobacterial genome diversification with relevance to antimicrobial resistance and host adaptation.

Genomic integrity is constantly threatened by DNA damage arising from endogenous cell metabolism and exogenous environmental factors. DNA lesions, when not rectified by dedicated repair systems, can persist, block DNA replication, and induce lethal fork collapse. "Translesion DNA Synthesis" (TLS) is an ubiquitous tolerance pathway by which the blocked replicative polymerase is transiently replaced by an alternative DNA polymerase that traverses the lesion[1]. In *E. coli*, DinB (Pol IV) and UmuDC (Pol V) are critical mediators of TLS[2,3]. In vitro, DinB and UmuDC share common biochemical characteristics that facilitate their in vivo function, including low fidelity, low processivity, lack of proofreading activity, and ability to bypass a variety of lesions[4–6]. In *E. coli*, the expression of *dinB* and *umuDC* is inducible by DNA damage through the SOS response pathway[7] and they respectively confer tolerance to alkylating agents and UV[8–10]. Because of the flexibility of their active site required for lesion bypass[11], TLS polymerases catalyze mutagenesis and play a key role in evolutionary

fitness[12] or antibiotic resistance in bacteria[13] and carcinogenesis in eukaryotes[14]. In *E. coli*, DinB and UmuDC are mutagenic, inducing substitution mutations as well as indels[15–20].

*Mycobacterium tuberculosis* (Mtb) is the causative agent of tuberculosis (TB), which kills 1.5 million people annually[21]. The major challenges impeding TB eradication efforts include the lack of short regimens of therapy, likely due to antibiotic tolerance mechanisms, and mutational antibiotic resistance[22], which is a substantial global health problem[21]. Mtb acquires antimicrobial resistance exclusively through chromosomal mutations, in contrast to the widespread mechanism of lateral gene transfer in other pathogens[23]. Human macrophages, the natural habitat of Mtb, expose the bacterium to diverse stresses, many of which directly damage DNA[24–28]. Mtb DNA repair pathways, in particular translesion polymerases, represent a promising and underexplored target for new TB drugs due to their role in survival within the host and in antimicrobial resistance[13]. However,

[1]Immunology Program, Sloan Kettering Institute, New York, NY 10065, USA. [2]Molecular Biology Program, Sloan-Kettering Institute, New York, NY 10065, USA. [3]Immunology and Microbial Pathogenesis Graduate Program, Weill Cornell Graduate School, 1300 York Avenue, New York, NY 10065, USA. ✉e-mail: glickmam@mskcc.org

the molecular pathways controlling chromosomal mutagenesis in mycobacteria are only partially understood. The replication fidelity of the Mtb chromosome is preserved by the proofreading function of the replicative polymerase DnaE1 that, when mutated, drastically increases mutation frequency[29]. Mycobacteria do not encode UmuDC but rather another TLS polymerase, a paralogue of DnaE1 called DnaE2[30,31]. In Mtb, *dnaE2* expression is dependent on DNA damage response[13,32]. DnaE2 is involved in UV tolerance as well as UV-induced mutagenesis and also contributes to bacterial pathogenicity and the emergence of drug resistance[13,33]. To date, DnaE2 is the only non-replicative polymerase known to contribute to chromosomal mutagenesis in mycobacteria.

Mycobacterial genomes encode several DinBs paralogs: two in Mtb (*dinB1*/*dinX*/Rv1537 and *dinB2*/*dinP*/Rv3056) and three in the nonpathogenic model *Mycobacterium smegmatis* (*dinB1*/MSMEG_3172, *dinB2*/MSMEG_2294/MSMEG_1014 and *dinB3*/MSMEG_6443)[30,34]. Whereas Mtb *dinB2* expression is enhanced by novobiocin[35], Mtb *dinB1* is part of the SigH regulon[36] and its expression is induced by rifampicin[35] and during human pulmonary TB infection[37]. In silico and experimental evidence indicates that DinB1, but not DinB2 nor DinB3, has a C-terminal β clamp binding motif and interacts directly with the β clamp in a heterologous organism[38]. *M. smegmatis* DinB1, DinB2, and DinB3 catalyze DNA-templated primer extension in vitro[39,40]. Initial characterization of a *dinB1dinB2* double mutant of Mtb, as well as the expression of the proteins in *M. smegmatis*, failed to identify a role in DNA damage tolerance, mutagenesis or pathogenicity in vivo[38].

Here we genetically investigate the contribution of mycobacterial TLS polymerases to DNA damage tolerance, antibiotic resistance, and mutagenesis. We show that DinB1 is highly mutagenic in vivo with a strong ability to incorporate substitution mutations conferring the resistance to rifampicin, one of the main drugs used to treat TB, and with a distinct mutagenic signature compared to DnaE2-catalyzed resistance mutations. We also demonstrate a previously unappreciated role for mycobacterial translesion polymerases in frameshift (FS) mutagenesis, with DinB1 and DnaE2 acting as the primary agents of spontaneous and UV-induced homo-oligonucleotide −1 FS mutagenesis in the mycobacterial chromosome.

## Results

### DinB1[Mtb] activity requires five N-terminal amino acids omitted from the annotated ORF

To investigate the role of DinB1 in mycobacteria, we expressed *M. smegmatis dinB1* (*dinB1*[Msm]) from an Anhydrotetracycline (ATc) inducible promoter (tet promoter). Addition of ATc enhanced the level of *dinB1*[Msm] mRNA in cells by 1000-fold (Supplementary Fig. 1a). We found that the expression of *dinB1*[Msm] in *M. smegmatis* caused a substantial growth defect and loss of viability (Fig. 1a, b) that was proportional to the level of inducer (Supplementary Fig. 1b, c). *dinB1*[Msm] expression also caused DNA damage, as evinced by an increase in the steady-state level of the RecA protein at 4 h post-induction by ATc (Fig. 1c).

The effects of expression of *M. smegmatis dinB1* contrast with the lack of similar findings when the Mtb gene was expressed[38], despite an overall identity of the two proteins of 75% (Supplementary Fig. 2). We found that expression of the Mtb *dinB1* gene (*dinB1*[Mtb]) in *M. smegmatis* did not phenocopy *dinB1*[Msm] (Fig. 1d). However, we reanalyzed the annotation of the *dinB1* open reading frames from *M. smegmatis* and Mtb and found an alternative translational start codon fifteen nucleotides upstream of the annotated start codon of Mtb *dinB1* used in prior experiments (Fig. 1e and Supplementary Fig. 2). Expression of the mRNA encoding this longer form of the Mtb DinB1 (*dinB1*[Mtb+5aa]) was induced 250-fold by ATc addition (Supplementary Fig. 1a) and impaired *M. smegmatis* growth (Fig. 1d), suggesting that the first five amino acids of DinB1 are essential for in vivo activity. These results indicate that prior conclusions about lack of in vivo activity of Mtb DinB1 are attributable to expression of a truncated protein.

A catalytic dead mutant of *dinB1*[Msm] (*dinB1*[D113A]), which is unable to catalyze DNA synthesis in vitro[40], exacerbated growth and viability defects compared to the WT polymerase (Fig. 1a, f). In contrast, expression of a *dinB1*[Msm] mutant lacking its β clamp binding domain (*dinB1*[Δβclamp]), predicted to not interact with the replicative machinery, did not cause growth inhibition or cell death (Fig. 1f, g). These results suggest that DinB1 interacts with the replicative machinery in vivo and competes with the replicative DNA polymerase at replication forks, as proposed in *E. coli*[41,42]. We cannot exclude the possibility that DinB1[ΔB-clamp] derivative is not as stable as WT and that the DinB1-dependent growth defect is unrelated to replication.

### DinB1 is an error-prone polymerase inducing antibiotic resistance through a characteristic mutagenic signature

Because of their active site flexibility required for lesion bypass, most translesion polymerases are error prone[11]. To determine the mutagenic capability of mycobacterial DinB1 as well as its ability to induce antibiotic resistance, we measured the frequency of rifampicin resistance (rif[R]), conferred by substitution mutations in the *rpoB* gene, in strains with temporally controlled expression of *dinB1*[Msm], *dinB1*[Mtb], or *dinB1*[Mtb+5aa]. In the strains carrying the empty vector or the *dinB1*[Mtb] plasmid, we respectively detected an average of 5.5 and 3.1 rif[R]/10[8] CFU 16 h after inducer addition (Fig. 2a). The expression of *dinB1*[Msm] or *dinB1*[Mtb+5aa] increased the frequency of rif[R] by six- or eight-fold but the expression of *dinB1*[Δβclamp] had no effect, suggesting that an interaction between DinB1 and the replicative machinery is required for DinB1-dependent mutagenesis. We observed a similar induction of mutagenesis after *dinB1*[Msm] expression in ΔrecA and ΔdnaE2 backgrounds (Supplementary Fig. 1d), showing that the effect of *dinB1* on mutation frequency is not the consequence of the RecA-dependent DNA damage response or the previously defined role of DnaE2 in mutagenesis[13], further strengthening the conclusion that DinB1 can be directly mutagenic.

To determine the mutation spectrum induced by DinB1, we sequenced the rifampicin resistance determining region (RRDR) of the *rpoB* gene (Fig. 2b). In absence of *dinB1* expression, the majority of RRDR mutations were either G>A or C>T (37%) or A>G or T>C (28%), with a minority of other mutations. Expression of *dinB1*[Msm] and *dinB1*[Mtb+5aa] strongly enhanced the relative proportion of A>G or T>C. The absolute frequency of these mutations was increased by 18- and 21-fold after *dinB1*[Msm] and *dinB1*[Mtb+5aa] expression, respectively. Around 75% of *rpoB* mutations found after *dinB1* expression were in the second nucleotide of the His442 codon compared with 25% in control cells (Fig. 2c and Supplementary Table 4). The mutation was almost exclusively CAC>CGC (His>Arg) and its absolute frequency was increased 21- and 28-fold after *dinB1*[Msm] and *dinB1*[Mtb+5aa] expression, respectively (Fig. 2d and Supplementary Table 4).

These results demonstrate that DinB1 is prone to induce mutations in vivo, with a characteristic mutagenic signature, A>G or T>C transition mutations, that contributes to rifampicin resistance at a specific codon in RpoB.

### DnaE2 but not DinBs mediates stress-induced substitution mutagenesis

The intrinsic mutagenicity of DinB1 demonstrated above supports a role for the enzyme in chromosomal mutagenesis at high copy number. We next measured the relative contributions of mycobacterial TLS polymerases (DinBs and DnaE2) in spontaneous mutagenesis by characterizing *M. smegmatis* cells lacking *dnaE2*, all *dinBs*, or all known translesion polymerases (*dnaE2 + dinBs*).

In the WT strain, we detected around 5 rif[R]/10[8] CFU (Fig. 3a) and 30% of *rpoB* mutations found in rif[R] colonies were A>G or T>C, 27% were G>A or C>T, and the remainder distributed across other mutation types (Fig. 3b). The deletion of TLS polymerases did not alter the frequency of rif[R] or shift the proportion of mutation types found in the

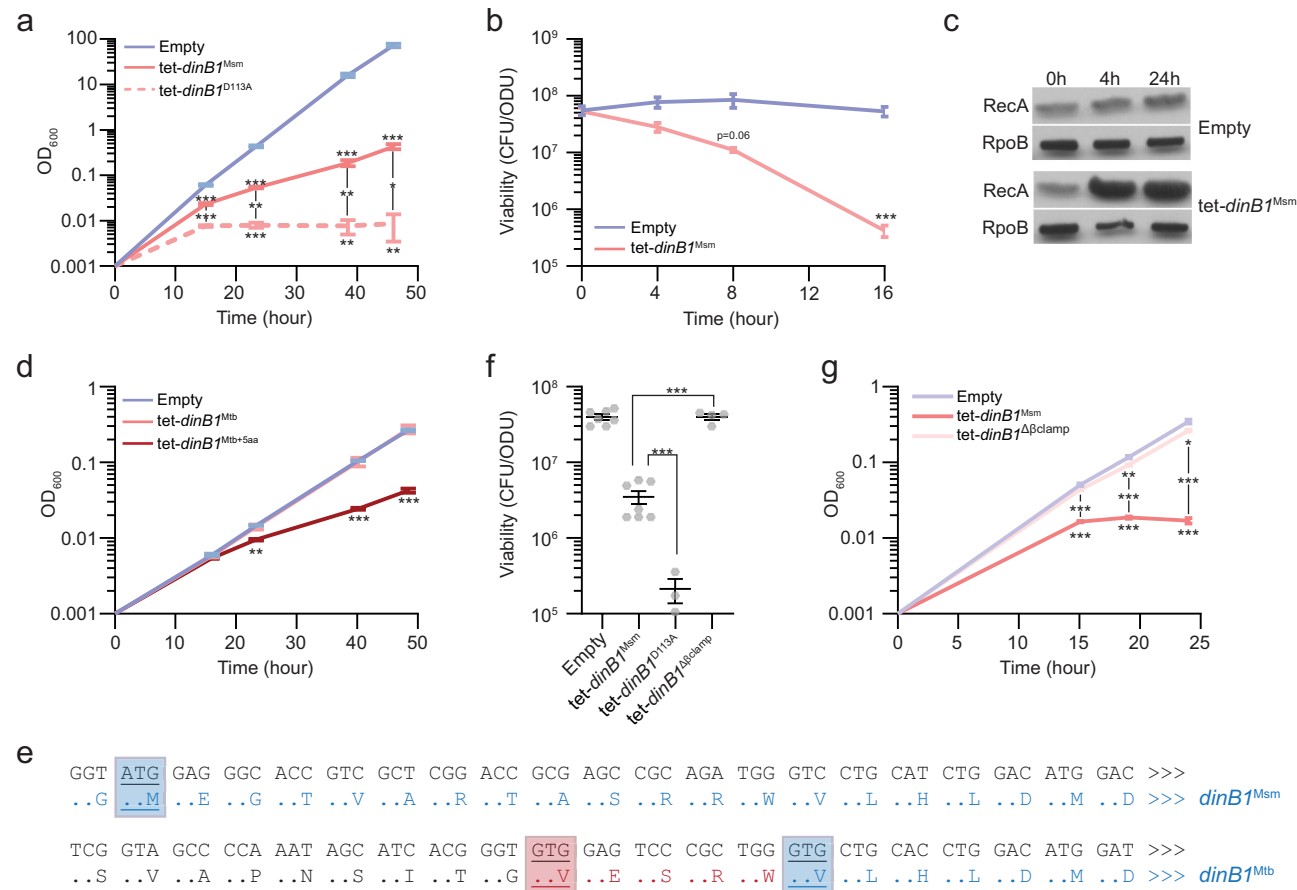

**Fig. 1 | DinB1^Mtb activity requires five N-terminal amino acids omitted from the annotated ORF. a**, **d**, **g** Growth and **b**, **f** viability of strains carrying an inducible (tet = Anhydrotetracycline inducible promoter) DinB1 or its indicated derivatives (Msm = *M. smegmatis*, Mtb = annotated *M. tuberculosis* DinB1, Mtb+5aa = N terminal extended DinB1, *dinB1* D113A = catalytically inactive *M. smegmatis* DinB1, Δβ clamp (ΔQESLF: 356–360 amino acids of *M. smegmatis* DinB1)) in presence of inducer. For **a**, the plotted OD value is the result of continuous dilution to maintain log phase growth (see methods). The viability in **f** was measured 24 h after inducer addition. **c** Anti-RecA/RpoB immunoblot from indicated strains with indicated times of inducer treatment. **e** Alignment of Msm and Mtb DinB1 N-termini with the potential start codons underlined. The blue boxed valine corresponds to the start codon of the published DinB1 noted as DinB1^Mtb above, whereas the red boxed valine shows an alternative start codon of an extended DinB1 denoted as DinB1^Mtb+5aa. Results shown are means (±SEM) of biological triplicates (**a**, **b**, **d**, **g**) or from biological replicates symbolized by gray dots (**f**). Stars above or under the means mark a statistical difference with the reference strain (empty vector) and lines connecting two strains show a statistical difference between them (*, $P < 0.05$; **, $P < 0.01$; ***, $P < 0.001$). *p* values were obtained on log-transformed data by one-way (**f**) or two-way (**a**, **b**, **d**, and **g**) ANOVA with a Bonferroni post-test.

*rpoB* gene (Fig. 3a, b), indicating that the activities of DnaE2 and DinBs are not the predominant mediators of substitution mutagenesis in the basal conditions tested.

By analyzing our recently published transcriptomic data[32], we found that the expression of *dinB1*, *dinB3*, and *dnaE2* was induced by UV and ciprofloxacin (cip) in *M. smegmatis* whereas *dinB2* expression was unaffected (Supplementary Fig. 3). The expression level of *dinB1* and *dinB2* in UV-irradiated cells was not impacted by *recA* deletion (Supplementary Fig. 3). By contrast, UV induction of *dinB3* and *dnaE2* expression was reduced in the Δ*recA* mutant, a result we confirmed by RT-qPCR (Supplementary Fig. 3).

We next investigated the relative contribution of DinBs and DnaE2 to stress-induced mutagenesis by measuring the frequency of rif^R in strains exposed to UV, hydrogen peroxide ($H_2O_2$) and methyl methanesulfonate (MMS). In the WT strain, we found 108-, 16-, and 24-fold increases of the rif^R frequency after treatment with UV, $H_2O_2$, and MMS, respectively (Fig. 3a). All three mutagens increased the relative and absolute frequencies of G>A or C>T mutations in the RRDR whereas UV also enhanced A>C or T>G mutations and $H_2O_2$ increased G>C or C>G mutations (Fig. 3b). Induction of the rif^R frequency by UV and $H_2O_2$ was DnaE2-dependent, declining by 10- and 4-fold in Δ*dnaE2* cells compared

to WT, whereas MMS-induced mutagenesis was not impacted (Fig. 3a). All types of UV- and $H_2O_2$-induced rif^R mutations (G>A or C>T, A>C or T>G and G>C or C>G) were reduced by the *dnaE2* deletion (Fig. 3b). In contrast, the *dinB123* deletion did not significantly decrease the mutation frequency in cells treated with UV, $H_2O_2$, and MMS or change the spectrum of mutation types (Fig. 3a, b). The analysis of *rpoB* mutations incorporated by DnaE2 during oxidative stress (WT vs Δ*dnaE2*) revealed that, unlike DinB1, DnaE2 conferred rifampicin resistance by mutating diverse *rpoB* codons (Fig. 3c and Supplementary Table 4). Particularly, we found that the presence of DnaE2 increased the absolute mutation frequency in Ser447(TCG>TTG), Ser438(TCG>TTG), His442(CAC>TAC), His442(CAC>GAC), and Asn435(AAC>AAG), but not the DinB1 associated mutation His442(CAC>CGC).

These results show that DnaE2, but not DinBs, contributes to UV- and $H_2O_2$-induced substitution mutagenesis with a distinct mutation spectrum from the DinB1 signature.

## Redundancy of DinB1 and DnaE2 in tolerance to alkylation damage and N²-dG adducts
The mutagenic properties of DinBs are intimately linked to their active site flexibility required in lesion bypass, a property that

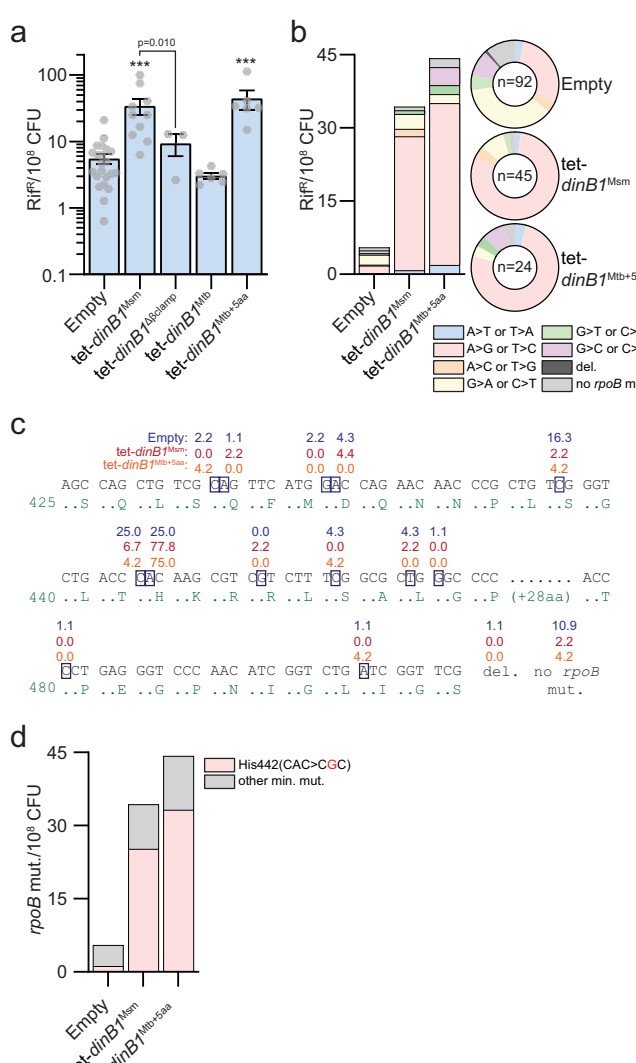

**Fig. 2 | DinB1 is an error-prone polymerase inducing antibiotic resistance through a characteristic mutagenic signature. a** Rifampicin resistance (rif$^R$) frequency in indicated strains in presence of inducer. Results shown are means (±SEM) of data obtained from biological replicates symbolized by gray dots. Stars above bars mark a statistical difference with the reference strain (empty vector) and lines connecting two strains show a statistical difference between them (***, $P < 0.001$). p-values were obtained on log-transformed data by one-way ANOVA with a Bonferroni post-test. **b** Relative (pie chart) and absolute (bar chart) frequencies of nucleotide changes detected in *rpoB* of rif$^R$ clones from the indicated strains. The number of sequenced rif$^R$ is given in the center of each pie chart. **c** Location and relative frequency in % of mutated nucleotides in *rpoB* found in empty (blue), tet-*dinB1*$^{Msm}$ (red) or tet-*dinB1*$^{Mtb+5aa}$ (orange) rif$^R$. **d** Absolute frequency of the main *rpoB* mutations found in indicated strains.

confers tolerance to agents that damage template DNA. We, therefore, investigated the role of mycobacterial TLS polymerases in DNA damage tolerance. A previous study did not identify a role for Mtb *dinB1* and *dinB2* in damage tolerance[38] but the possibility of redundancy between *dinB*s and *dnaE2* was not tested. The *M. smegmatis ΔdnaE2dinB123* mutant was not more sensitive than the WT to H$_2$O$_2$ (Supplementary Fig. 4a) or ciprofloxacin (Supplementary Fig. 4b). As reported by Boshoff et al.[13], we found that the *ΔdnaE2* deletion increased the sensitivity to UV (Supplementary Fig. 4c) and mitomycin C (Supplementary Fig. 4d) but we did not observe an additive effect caused by *ΔdinB123* deletion.

We next investigated the role of DnaE2 and DinBs in the tolerance to alkylation damage, as reported for *E. coli* DinB[8,10], by testing the sensitivity of *M. smegmatis* TLS polymerase mutants to the chemical methylating agents MMS and methylnitronitrosoguanidine (MNNG). Using disc diffusion assays, we found that the *ΔdinB123* or *ΔdnaE2* mutants were not more sensitive than the WT strain to MMS but the loss of the four TLS polymerases in combination conferred higher sensitivity (Fig. 4a), indicating substantial redundancy. We obtained similar results by plating serial dilutions of these strains on agar medium containing MMS (Supplementary Fig. 4f) and with disc diffusion using MNNG (Fig. 4c). The *ΔdnaE2dinB123* MMS and MNNG sensitivity was partially complemented by the introduction of an ectopic copy of *dnaE2*$^{Msm}$ or *dinB1*$^{Msm}$ but not *dinB2*$^{Msm}$ or *dinB3*$^{Msm}$ (Fig. 4b, d and Supplementary Fig. 4f, g). The *ΔdnaE2dinB123* sensitivity to MMS and MNNG was partially complemented by an ectopic copy of *dinB1*$^{Mtb}$ but not *dinB2*$^{Mtb}$ (Fig. 4b, d), indicating conservation of DinB1 activities between fast- and slow-growing mycobacteria.

Finally, we measured the effect of loss of *dinB*s and *dnaE2* on tolerance to N$^2$-dG adducts. None of the translesion polymerase mutants was impacted in their tolerance to 4-nitroquinoline-1oxide (4-NQO) (Supplementary Fig. 4e). Whereas the *ΔdinB1*, *ΔdinB2*, and *ΔdnaE2* single mutants were slightly more sensitive than the WT strain to nitrofurazone (NFZ)[10], the *ΔdinB123* and *ΔdnaE2dinB123* mutants were highly sensitive (Fig. 4e). Ectopic expression of *dinB1*$^{Msm}$, *dinB3*$^{Msm}$, *dnaE2*$^{Msm}$, or *dinB1*$^{Mtb}$ in the *ΔdnaE2dinB123* strain partially reversed the NFZ sensitivity (Fig. 4f), reinforcing the substantial redundancy of translesion polymerases in mycobacteria for bypassing damage. These results reveal previously unrecognized roles for DnaE2 and DinB1 in the tolerance to genomic alkylation damage and N$^2$-dG adducts in mycobacteria and suggest a dominant role of DinB1 over the other mycobacterial DinBs in TLS.

### DinB1 mediates −1 frameshift mutations

The data above indicate that DinB1 catalyzes substitution mutations that confer resistance to antibiotics such as rifampicin by abolishing drug binding while maintaining the functionality of the essential antibiotic target. However, the diversity of mutational alterations of the chromosome that impact bacterial phenotypes includes not only substitutions, but chromosomal rearrangements, deletions, and FS mutations. Recently, FS mutagenesis has emerged as an important mechanism of genome diversification in mycobacteria[43,44] but the agents of FS mutagenesis in mycobacteria are not known.

Translesion polymerases can introduce FS mutations during lesion bypass[1–3], which prompted us to query the role of DinBs and DnaE2 in FS mutagenesis. To detect −1 FS mutations, we created a reporter system in which the chromosomal *leuD* gene carries a 2-base pair deletion in the second codon (*leuD*$^{-2}$), which confers leucine auxotrophy (Fig. 5a). Reversion of this mutation by −1 or +2 FS confers leucine prototrophy (leu$^+$) which is selected on leucine free media. In WT cells, the reversion frequency was 5 leu$^+$/10$^8$ CFU (Fig. 5b). Sequencing of *leuD* in leu$^+$ colonies revealed a −1 deletion in a run of 3T in almost half the revertants (44%). The other half had +2 addition (10%), >2-nucleotide insertion (7%), >2-nucleotide deletion (5%), or no mutation in *leuD* (32%) (Fig. 5b and Supplementary Table 5). The expression of the inactive *dinB1*$^{Mtb}$ did not increase leu$^+$ frequency, but expression of *dinB1*$^{Msm}$ or *dinB1*$^{Mtb+5aa}$ increased leu$^+$ frequency 4- or 27-fold due to a dominant proportion of −1 FS in the homo-oligonucleotide run of 3T.

We then investigated the ability of DinB1 to incorporate +1 FS mutations using a *leuD* reporter with a one nucleotide deletion in the second codon (*leuD*$^{-1}$) (Supplementary Fig. 5a). In WT cells, the leu$^+$ frequency was 1/10$^7$ CFU but the mutations were mixed between +1 FS in *leuD* and an unexpected class of −1 FS mutations at the 3′ end of the upstream *leuC* gene, which suppressed the native *leuC* stop codon and restored the reading frame of the *leuD* coding sequence to create a

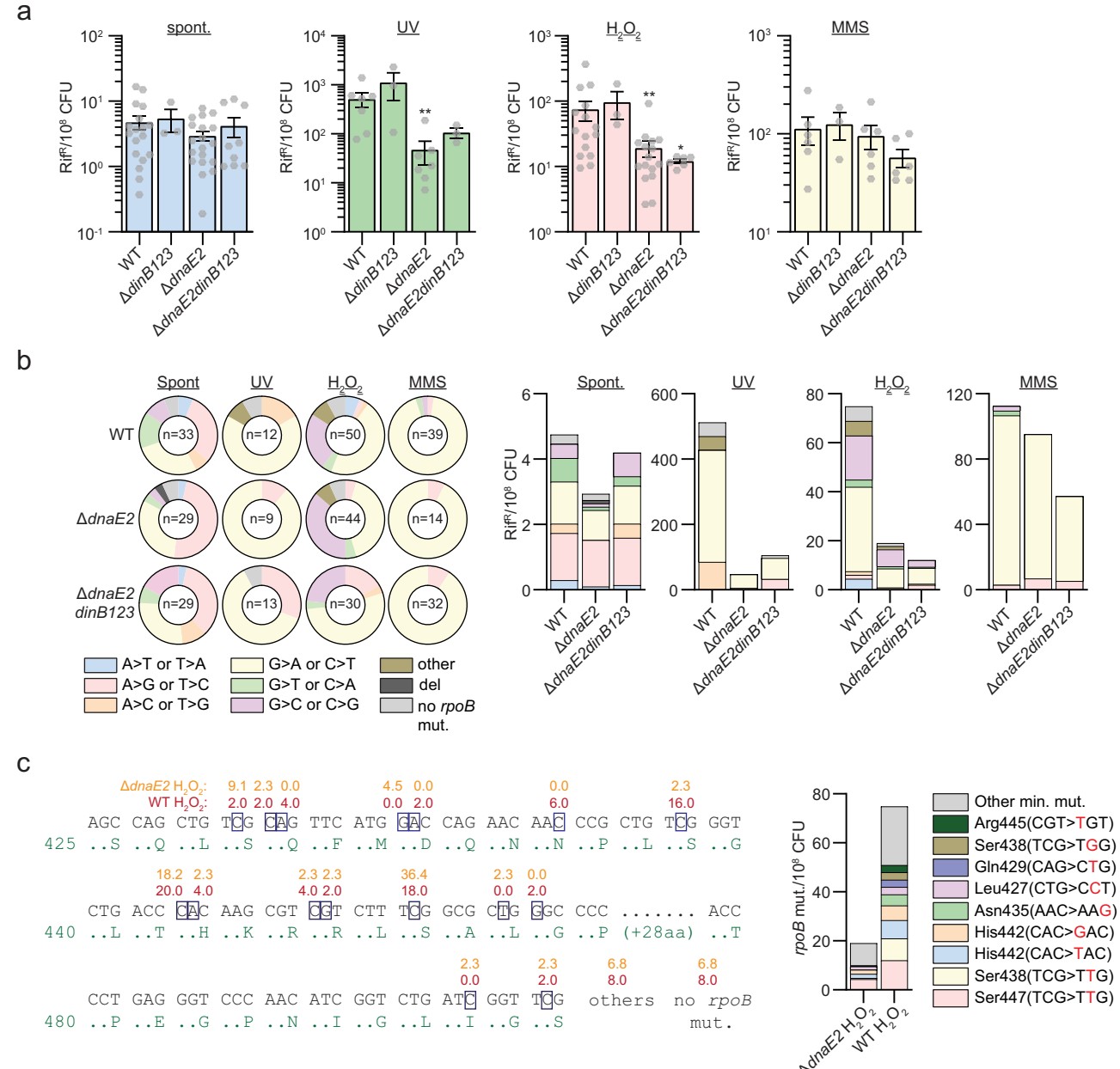

**Fig. 3 | DnaE2 but not DinBs mediates stress-induced substitution mutagenesis.**
**a** Rifampicin resistance (rif^R) frequency in indicated strains and conditions. Results shown are means (±SEM) of data obtained from biological replicates symbolized by gray dots. Stars above bars mark a statistical difference with the reference strain (WT of each condition) (*, $P < 0.05$; **, $P < 0.01$). p-values were obtained on log-transformed data by one-way ANOVA with a Bonferroni post-test. **b** Relative (pie chart) and absolute (bar chart) frequencies of nucleotide changes detected in *rpoB* of rif^R clones from the indicated strains. The number of sequenced rif^R is given in the center of each pie chart. **c** Location and relative frequency of mutated nucleotides of *rpoB* found in rif^R of ΔdnaE2 + H₂O₂ (orange) or WT + H₂O₂ (red). The bar chart shows the absolute frequency of the main *rpoB* mutations found in indicated strains.

LeuC-LeuD fusion protein (Supplementary Fig. 5a). The expression of *dinB1^Msm* increased leu⁺ frequency by 16-fold but the sequencing of leu⁺ colonies revealed that *dinB1* expression exclusively catalyzed −1 FS at the 3′ end of the *leuC* gene, with 97% of these mutations in a 3C run. These results reveal that DinB1 can promote −1 FS mutations in the mycobacterial genome and does so more efficiently than promoting +1 FS mutations.

### DinB1 incorporates −1 FS and +1 FS in homo-oligonucleotide runs in vivo

The location of the FS mutations in short homo-oligonucleotide tracts of *leuC* and *leuD* suggested that DinB1 may be a catalyst of FS mutagenesis in low complexity sequences. To more precisely measure the capability of DinB1 to incorporate −1 and +1 FS in homo-oligonucleotide runs in vivo and determine the effect of homomeric template sequence, we constructed integrative plasmids carrying a gene which confers resistance to kanamycin (*kan*) inactivated by the incorporation of 4T (*kan*::4T), 4C (*kan*::4C), 4G (*kan*::4G), 4A (*kan*::4A), 5T (*kan*::5T), 5G (*kan*::5G), or 5A (*kan*::5A) runs in the coding strand immediately 3′ of the start codon (Fig. 5a). These plasmids do not confer kanamycin resistance, but a deletion of one nucleotide (−1 FS) in the 4N run or +1 FS in the 5N run will restore the reading frame of *kan* allowing selection for kan^R.

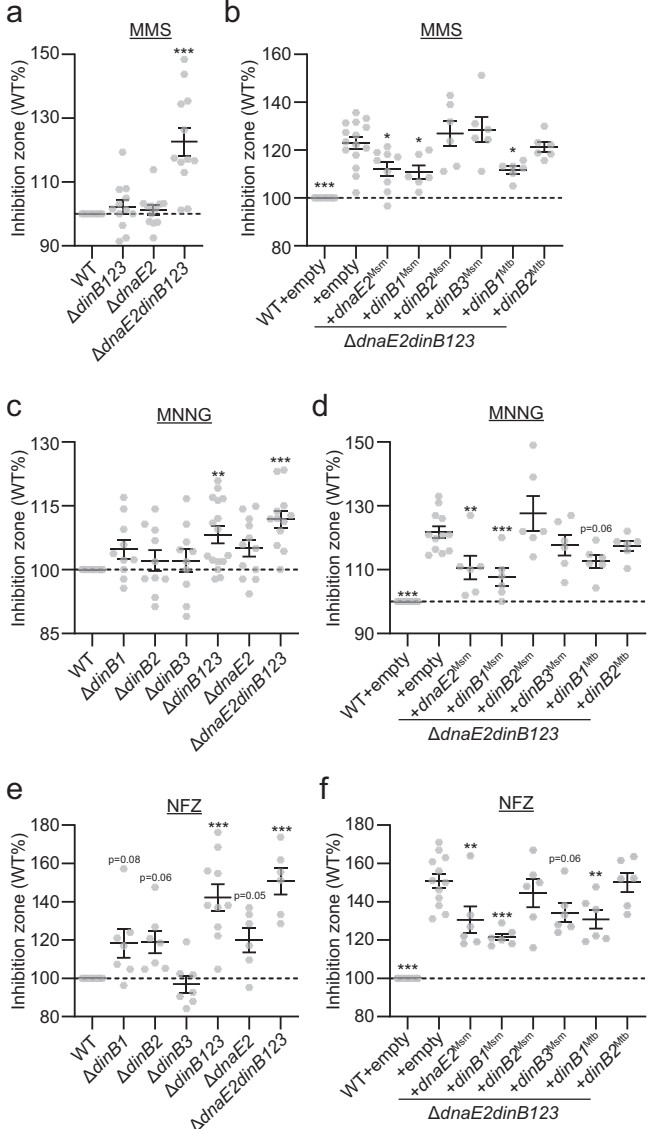

**Fig. 4 | Redundancy of DinB1 and DnaE2 in tolerance to alkylation damage and N$^2$-dG adducts.** Sensitivities of indicated strains to **a** and **b** MMS, **c** and **d** MNNG, or **e** and **f** NFZ were measured by disc diffusion assay. Translesion polymerases were expressed under their native promoter from a genome integrated vector in panels **b**, **d**, and **f**. Results shown are means (±SEM) of data obtained from biological replicates symbolized by gray dots. Stars above the means mark a statistical difference with the reference strain (WT or Δ*dnaE2dinB123* + empty in complementation experiments) (*, $P < 0.05$; **, $P < 0.01$; ***, $P < 0.001$). *p* values were obtained on log-transformed data by one-way ANOVA with a Bonferroni post-test.

We first measured the ability of DinB1 to incorporate −1 FS in homo-oligonucleotide runs by using the *kan*::4N constructs. We found between 5 and 18 kan$^R$/10$^8$ CFU, depending on the run sequence, in strains carrying the empty vector and the vast majority kan$^R$ colonies had −1 FS in the homo-oligonucleotide runs (Fig. 5c and Supplementary Fig. 5b, c, d). Expression of *dinB1*$^{Msm}$ enhanced −1 FS in runs of 4T, 4C, 4G, and 4A by 12-, 19-, 11-, and 3-fold, respectively. *dinB1*$^{Mtb+5aa}$ expression also increased −1 FS by 14- and 19-fold in 4T and 4C runs, respectively, but had no effect on runs of 4G or 4A.

By using the *kan*::5N constructs, we quantified the capacity of DinB1 to incorporate +1 FS in homo-oligonucleotide runs. Between 10 and 74 kan$^R$/10$^8$ CFU were detected in strains carrying the empty vector, depending on the nature of the run (Fig. 5d and Supplementary 5e, f). We found +1 FS in the run of the majority of the sequenced

kan$^R$ colonies of all strains. Expression of *dinB1* increased the frequency of +1 FS localized in runs of 5T, 5G, and 5A by 3-, 6-, and 21-fold, respectively. *dinB1*$^{Mtb+5aa}$ expression also elicited +1 FS in the 5T (10-fold increase), 5G (3-fold increase), and 5A (5-fold increase) runs. Overall, these data reveal that mycobacterial DinB1 is a strong mediator of −1 FS and +1 FS in homo-oligonucleotide runs.

### DinB1 can slip on homo-oligonucleotide runs in vitro

To test if the DinB1 polymerase is prone to slippage on homo-oligomeric template tracts in vitro, we employed a series of 5′ $^{32}$P-labeled primer-template DNA substrates consisting of a 13-bp duplex with a 5′-tail composed of a run of four, six, or eight consecutive A nucleotides (A4, A6, A8) immediately following the primer 3′-OH terminus and flanked by three C nucleotides (Fig. 5e). Reaction of a DNA polymerase with the A4, A6, and A8 primer-templates in the presence of only dTTP should, if the polymerase does not slip or misincorporate dTMP opposite the template nucleotide following the A run, allow for the addition of four, six, or eight dTMP nucleotides to the primer terminus. However, if the polymerase is prone to backward slippage, then the primer strand can recess and realign to the template to allow one or more additional cycles of dTMP addition. Whereas most of the primer extension events catalyzed by DinB1 on the A4, A6, and A8 templates in the presence of dTTP did indeed cease at the end of the A run (e.g., denoted by ▶ for the A4 reaction in Fig. 5e), we consistently detected the synthesis of a minority product one nucleotide longer, consistent with a single slippage step mimetic of a + 1 frameshift (Fig. 5e). DinB1 displayed similar behavior when reacted with a series of DNAs in which the template strand tail comprised a run of four, six, or eight consecutive T nucleotides (T4, T6, T8) followed by three G nucleotides (Fig. 5e). In presence of dATP, the majority of the primer extension events on the T4, T6, and T8 templates entailed 4, 6, and 8 cycles of dAMP addition, respectively. +1 slippage products were also detected. The propensity to slip, defined as +1/[+1 plus ▶], increased progressively as the template T tract was lengthened from T4 (1%) to T6 (11%) to T8 (16%) (Fig. 5e). The +1 products are unlikely to have arisen via addition of a 3′-terminal mismatched dNMP, insofar as we could detect no extension of the 13-mer primer stands on the A6 and T6 templates when DinB1 was presented with the incorrect dNTP (Supplementary Fig. 6).

The finding that a DinB1 is capable of +1 slippage synthesis on a homo-oligomeric tract when the only dNTP available is that templated by the homo-oligomer does not reflect the situation in vivo where the polymerase will have access to the next correctly templated dNTP. To attempt to query whether provision of the next templated dNTP in vitro suppresses slippage, we included a dideoxy NTP (ddNTP): either ddGTP templated by the run of three C nucleotides following the A4, A6, and A8 tracts or ddCTP templated by the run of three G nucleotides flanking the T4, T6, and T8 tracts. ddNTPs are employed to force termination upon incorporation of the first templated nucleotide following the homo-oligomeric tract. Inclusion of the next templated ddNTP following the homo-oligomeric template tract suppressed the single slippage step mimetic of a + 1 frameshift observed in absence of ddNTPs (Fig. 5e), suggesting that +1 slippage by DinB1 is dampened by the presence of the next correctly templated dNTP.

DinB1 was, in presence of ddNTPs, only partially effective in triggering conversion of the 17, 19, and 21 nt species to the respective ddG- or ddC-terminated 18, 20, and 22 nt products (Fig. 5e). We considered several possibilities, including: (i) that DinB1 might disengage from the primer-template when the 5′ tail comprising the template strand becomes shorter, and hence lose efficiency in adding opposite the third nucleotide from the 5′ end of the template strand; or (ii) DinB1 is inherently feeble at using dideoxy NTPs as substrates; or (iii) DinB1 and the primer 3′-OH end can slip forward on the template run by a single nucleotide on the homo-oligonucleotide run

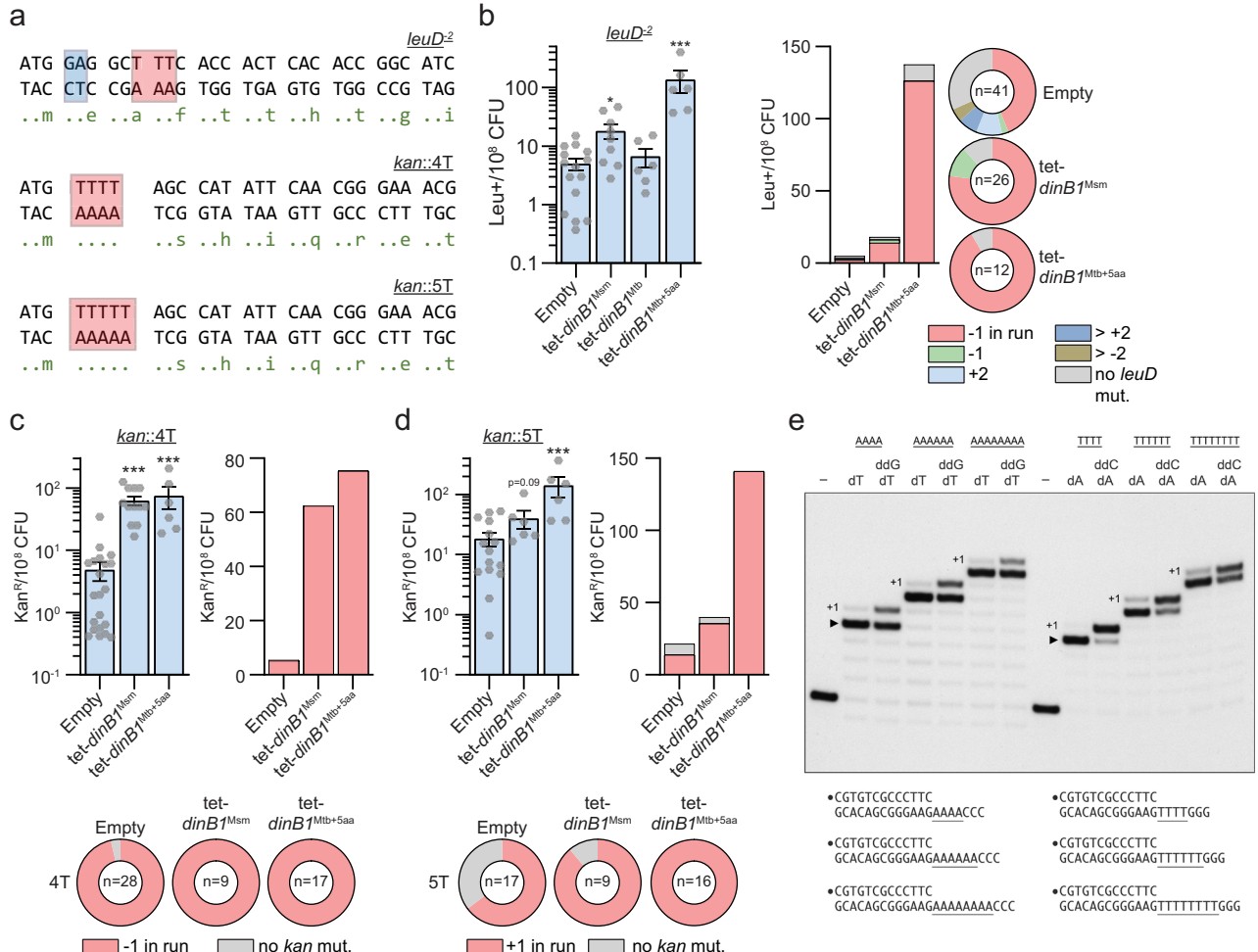

**Fig. 5 | DinB1 promotes −1 and +1 frameshift mutations in homo-oligonucleotide runs. a** *leuD* and *kan* frameshift (FS) reporter assays. *leuD* and *kan* open reading frame N-termini in which 2 base pairs in the second *leuD* codon were removed (blue box) or 4T/5T runs (red box) were incorporated upstream the start codon of *kan*. Reversion can occur by FS mutations that restore the *leuD* or *kan* reading frames resulting in phenotypic leucine prototrophy (leu⁺) or kanamycin resistance (kanᴿ). The red box in *leuD* shows the run of 3T in which the majority of detected FS in leu⁺ were found. **b** leu⁺ and **c, d** kanᴿ frequencies in the indicated strains in presence of inducer. Results shown are means (±SEM) of data obtained from biological replicates symbolized by gray dots. Stars above the means mark a statistical difference with the reference strain (empty) (*, *P* < 0.05; ***, *P* < 0.001). p-values were obtained on log-transformed data by one-way ANOVA with a Bonferroni post-test. Relative (pie chart) and absolute (bar chart) frequencies of nucleotide changes detected in *leuD* of leu⁺ cells or in *kan* of kanᴿ cells represented with colors: red = −1 FS in the **b** 3 T run, **c** 4 T run, or **d** 5 T run, other colors = FS outside of the run, and gray = no detected mutation. The number of sequenced leu⁺ or kanᴿ colonies is given in the center of each pie chart. **e** DinB1 polymerase reaction mixtures containing 5′ ³²P-labeled primer-template DNAs with A4, A6, A8, T4, T6, or T8 runs in the template strand (depicted below and included as indicated above the lanes) and 125 μM deoxynucleotides and dideoxynucleotides (as specified above the lanes) were incubated at 37 °C for 15 min. DinB1 was omitted from reactions in lanes −. The reaction products were analyzed by urea-PAGE and visualized by autoradiography. The +1 slippage products are indicated.

(mimetic of a −1 FS) and this species is extended by ddNMP incorporation to yield a product that comigrates with the 17, 19, and 21 nt species. These issues were addressed by reacting DinB1 with the A6 and T6 primer templates in the presence of various nucleotide substrates and combinations thereof. DinB1 catalyzed six steps of dTMP addition to the A6 template in the presence of dTTP and inclusion of dGTP elicited three further steps of dGMP addition opposite the 5′-terminal CCC element of the template strand (Supplementary Fig. 6), indicating that DinB1 is competent for fill-in synthesis. In the reaction with ddTTP only, a small fraction of the primer was elongated by the expected single nucleotide step showing that DinB1 is, as hypothesized in scenario (ii), unable to efficiently utilize ddTTP as a substrate for correct templated addition. Similar results were obtained for the T6 primer template (Supplementary Fig. 6). The question remains whether any of the residual 17, 19, and 21 nt species seen in Fig. 5e represent −1 slips consistent with scenario (iii) above. In the case of the T tract templates, we see that the fraction of products that fail to

be extended in the presence of ddCTP, defined as unextended product/[unextended product plus extended product], increases progressively as the template tract lengthens from 4T (13% unextended) to 6T (34%) to 8T (48%) (Fig. 5e). This result suggests a contribution of −1 slippage (rather than pure failure to incorporate ddC), the reasoning being that lengthening the template homo-oligonucleotide run is expected to enhance slippage but not impact DinB1's ability to incorporate ddNTPs.

## DinB1 is the primary mediator of spontaneous −1 FS in runs of homo-oligonucleotides

To determine the relative contribution of TLS polymerases in spontaneous FS mutagenesis, we measured the frequency of −1 FS in strains lacking *dnaE2* or *dinB*s using the *leuD*⁻² reporter (Fig. 6a). The Δ*dinB2*, Δ*dinB3*, and Δ*dnaE2* deletions did not reduce the leu⁺ frequency or the proportion of −1 FS detected in leu⁺ colonies (Fig. 6a and Supplementary Table 5). In contrast, the leu⁺ frequency decreased by 5-, 3-,

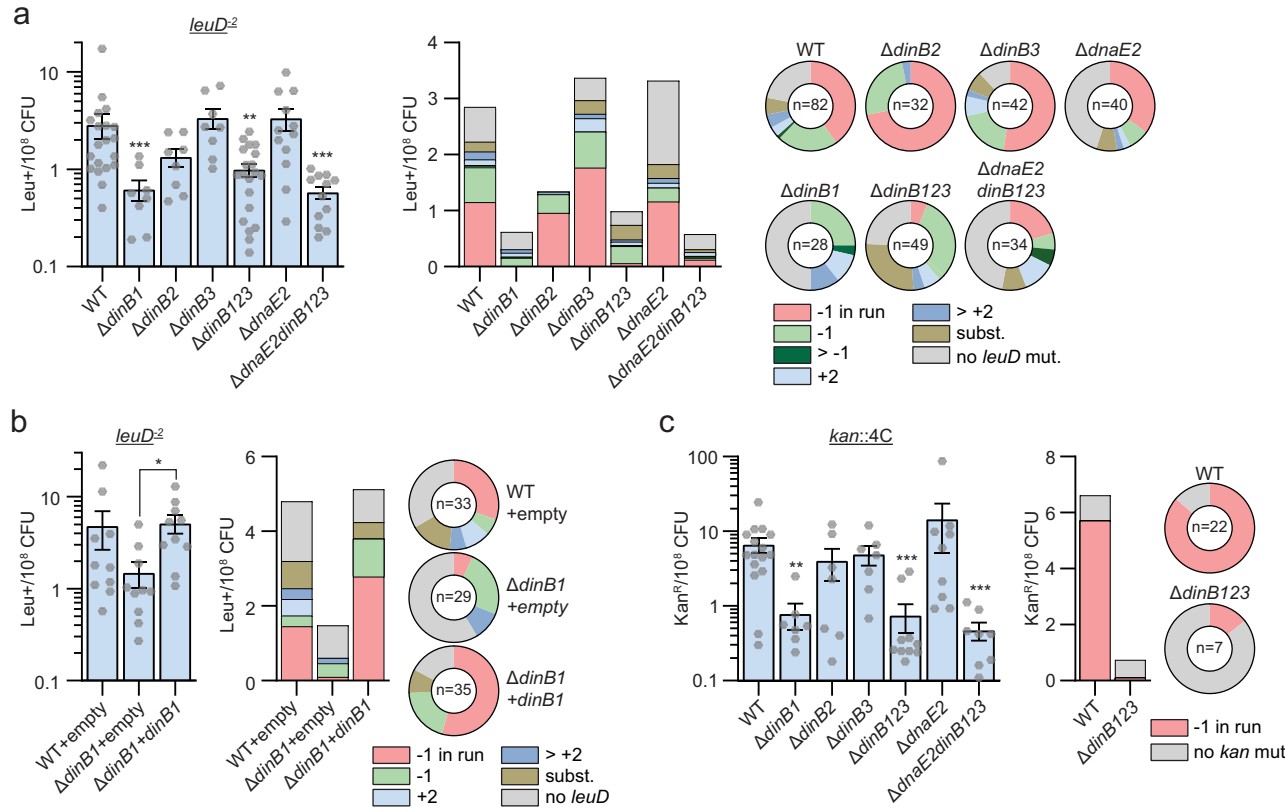

**Fig. 6 | DinB1 is the primary mediator of spontaneous −1 frameshift mutations in homo-oligonucleotide runs. a, b** leu⁺ or **c** kanᴿ frequency in the indicated strains and relative (pie chart) or absolute (bar chart) frequencies of nucleotide changes detected in *leuD* or *kan* coded by color. The number of sequenced leu⁺ or kanᴿ colonies is given in the center of each pie chart. Results shown are means (± SEM) of data obtained from biological replicates symbolized by gray dots. Stars above bars mark a statistical difference with the reference strain (WT) and lines connecting two strains show a statistical difference between them (*, *P* < 0.05; **, *P* < 0.01; ***, *P* < 0.001). p-values were obtained on log-transformed data by one-way ANOVA with a Bonferroni post-test.

and 5-fold in the Δ*dinB1*, Δ*dinB123* and Δ*dnaE2dinB123*, respectively, and the proportion of −1 FS mutations localized in the 3T run of *leuD* was also reduced. Expression of *dinB1* in the Δ*dinB1* strain restored both the WT leu⁺ frequency and the proportion of −1 FS localized in the 3T run of *leuD* (Fig. 6b and Supplementary Table 5).

We extended these findings using the *kan*::4N reporters (Fig. 6c and Supplementary 7a, b, c). Compared to WT, the kanᴿ frequency measured with the *kan*::4C reporter decreased by 9-, 9-, and 14-fold in the Δ*dinB1*, Δ*dinB123*, and Δ*dnaE2dinB123* mutants, respectively, but we did not detect decrement in the Δ*dinB2*, Δ*dinB3*, and Δ*dnaE2* mutants (Fig. 6c). The absolute frequency of −1 FS detected in the 4C run was reduced 54-fold in the Δ*dinB123* mutant. In contrast, there was no impact of *dinB1* deletion on the frequency of −1 FS mutation in 4T, 4G, or 4A runs (Supplementary Fig. 7a, b, c). In Δ*dinB123* cells, −1 FS in the 4T run was reduced 4-fold but was unaffected in the 4G and 4A runs. Finally, we did not detect a significant impact of the TLS polymerase deletions on spontaneous +1 FS mutagenesis using the *kan*::5T, *kan*::5G or *kan*::5A reporters (Supplementary Fig. 7d, e, f).

These results show that: (i) DinB1 is the dominant TLS polymerase involved in spontaneous −1 FS mutations in some homo-oligonucleotide runs; (ii) there is a redundancy between DinB1 and at least one other DinB for certain homopolymeric sequences; and (iii) endogenous levels of DinBs do not mediate spontaneous +1 FS mutations in unstressed cells.

**DnaE2 is the primary mediator of UV-induced −1 FS mutagenesis**

Prior literature[13] as well as our data above (Supplementary Fig. 3) show that some mycobacterial TLS polymerases are DNA damage inducible,

indicating that FS mutagenesis may be enhanced by DNA damage. To query the role of DNA damage in stimulating FS mutagenesis and the role of TLS polymerases in this process, we used the *leuD* and *kan* systems in conjunction with UV treatment. UV irradiation increased the frequency of leu⁺ in the WT strain carrying the *leuD*⁻² reporter by 9-fold due to the induction of three main mutation types: −1 FS in the 3T run, −1 FS outside of the run, and base substitutions that in many cases created a new in-frame translational start codon that restored LeuD (Fig. 7a and Supplementary Table 5). The induction of the three mutation types was reduced in Δ*dnaE2* and Δ*dnaE2dinB123* mutants. The residual UV-dependent increase of the −1 FS mutations in the *leuD* 3T run in the Δ*dnaE2* mutant was completely abolished in the Δ*dnaE2dinB123* mutant, suggesting redundancy between DnaE2 and DinBs for damage-induced frameshifting.

We also measured the impact of UV on −1 FS using the *kan* reporters. In the WT strain carrying the *kan*::4T vector, irradiation increased the frequency of kanᴿ by fivefold and 100% of the sequenced clones had a −1 FS in the homo-oligonucleotide run (Fig. 7b). The −1 FS frequency was not reduced in either the Δ*dnaE2* or Δ*dinB123* mutants but decreased fivefold in the Δ*dnaE2dinB123* mutant. UV treatment also enhanced −1 FS in the 4C, 4G, and 4A runs in WT cells but not in the *dnaE2* mutant (Supplementary Fig. 8a, b, c). 25% and 43% of the DnaE2-dependent mutagenic events detected with the *kan*::4G and *kan*::4A reporters, respectively, comprised −1 FS mutations located outside of the homo-oligonucleotide runs (Supplementary Fig. 8b, c), showing that the frameshifting activity of DnaE2 is not restricted to homo-oligonucleotide runs. Finally, although we detected a DnaE2-dependent increase of kanᴿ frequency in strains carrying *kan*::5T, *kan*::5G. or *kan*::5A reporters (Supplementary Fig. 8d, e, f), these events

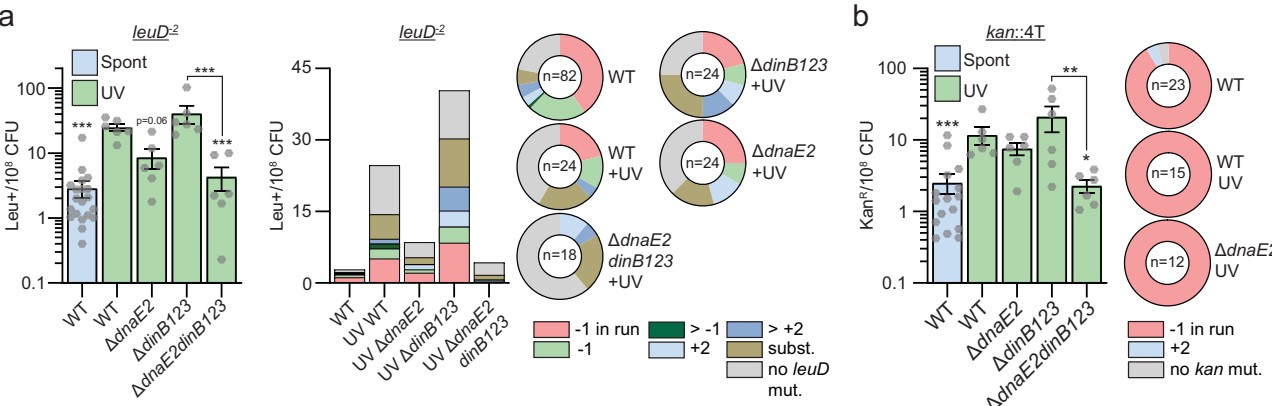

**Fig. 7 | DnaE2 is the primary mediator of DNA damage-induced −1 frameshift mutations in homo-oligonucleotide runs. a** leu+ or **b** kan^R frequency in indicated strains/conditions and relative (pie chart) or absolute (bar chart) frequencies of nucleotide changes detected in *leuD* or *kan* coded by color. The number of sequenced leu+ or kan^R colonies is given in the center of each pie chart. Results shown are means (±SEM) of data obtained from biological replicates symbolized by gray dots. Stars above bars mark a statistical difference with the reference strain (WT + UV) and lines connecting two strains show a statistical difference between them (*, P < 0.05; **, P < 0.01; ***, P < 0.001). p values were obtained on log-transformed data by one-way ANOVA with a Bonferroni post-test.

were not due to FS mutagenesis and rather were nucleotide substitutions that created a new start codon.

These results reveal that DnaE2 is a major contributor to −1 FS mutations in response to DNA damage and that these FS are not restricted to homo-oligonucleotide runs. We also found that the DinBs have a redundant role with DnaE2 in −1 FS mutagenesis in response to DNA damage which differs depending on the sequence context.

## Discussion

Two decades ago, Boshoff et al. highlighted the importance of DnaE2 in mutagenesis, DNA damage tolerance, and pathogenicity in Mtb[13]. Because prior attempts to deduce a function for DinBs failed to reveal any phenotype[38], DnaE2 has been considered the only mycobacterial TLS polymerase mediating chromosomal mutagenesis. However, our studies support a significant revision to this view and implicate a network of translesion polymerases in chromosomal mutagenesis.

We confirmed that DnaE2 catalyzes the acquisition of rifampicin resistance in response to DNA damage[13]. We found that DnaE2 has the ability to induce a spectrum of *rpoB* mutations, particularly S447L, S438L, H442Y, and H442D. The codons of these amino acids are conserved between *M. smegmatis* and Mtb and these mutations represent ~75% of the *rpoB* mutations detected in the sequenced rif^R Mtb clinical isolates (Supplementary Tables 4, 6, and[45]).

Here we report that DinB1 can induce rifampicin resistance through a mutagenic activity. In contrast to the diverse mutation spectrum of DnaE2, DinB1 confers rif^R through a unique *rpoB* mutation (CAC>CGC; H442R). This mutation has been detected in rif^R Mtb clinical isolates at a frequency between 0.8% and 5%, depending on the study[45–51] (Supplementary Tables 4, 6). Our data indicate shared and sequence context specific roles for DinB1 and DnaE2 in substitution mutagenesis and antibiotic resistance. This shared role raises several questions about how DinB1 and DnaE2 cooperate or compete at the replication fork. Both DinB1 and DnaE2 interact with the β clamp, DinB1 directly and DnaE2 via the ImuB protein[33,52]. Whether these two proteins compete for the same binding site, are recruited based on the type of damage, or can occupy different sites on the β clamp remains to be determined. Our study shows that DinB1 confers rifampicin resistance when the gene is highly expressed. Transcriptomic studies reported a 4.5-fold induction of *dinB1*^Mtb caused by rifampicin treatment of Mtb[35] and a 12-fold induction of the gene in pulmonary TB compared to culture conditions[37]. Further experiments will be needed to test if DinB1 participates to rifampicin resistance acquisition during infection.

This work, together with previous DnaE2 studies, suggests that TLS polymerases may be attractive drug targets to prevent the acquisition of antibiotic resistance in Mtb[13,33]. However, we recently showed that mycobacterial TLS polymerases contribute to antibiotic bactericidal action elicited by the genomic incorporation of oxidized nucleotides when the MutT system is depleted[53]. Thus, TLS inhibition might have salutary effects on resistance, while diminishing the bactericidal effects of some antibiotics, a balance that will need to be assessed.

In addition to its role in substitution mutagenesis, we found that DinB1 is the primary mediator of spontaneous −1 FS mutagenesis whereas DnaE2 is involved in DNA damage-induced −1 FS mutagenesis. We demonstrated that DinB1 is prone to FS in homo-oligonucleotide runs and that the FS mutagenic activity of DinB1 is conserved between *M. smegmatis* and Mtb. We observed low frequency of DinB1 slippage on homo-oligonucleotide runs in vitro, contrasting with the significant increase of −1 and +1 FS frequency measured in vivo when DinB1 is expressed. Because eukaryal translesion polymerase fidelity can be impacted by the PCNA clamp[54] and because we found here that DinB1 mutagenesis depends on its β clamp interaction, we speculate that slippage might be increased if the β clamp was tethering DinB1 to the template.

FS mutations are an increasingly recognized source of genomic diversity in Mtb. Mycobacterial genomes have a similar frequency of nucleotide substitutions compared to other bacteria but a higher frequency of FS mutations in homo-oligonucleotide runs[55]. This can be explained by the absence of the conventional MutS/L mismatch repair in mycobacteria, functionally replaced by a NucS-dependent system which can correct substitution mutations but not indels[56,57]. A recent in silico analysis of 5977 clinical Mtb isolates established that indels appear every 74,497 bases in genic regions and that the most common indel is −1 FS (two times more frequent than +1 FS and 6-fold more frequent than any other indel)[43]. These indels are significantly enriched in homo-oligonucleotide runs.

Two seminal recent studies[44,58] demonstrated a link between Mtb antibiotic tolerance and homo-oligonucleotide FS mutagenesis. Specifically, FS mutations in a run of 7C in the *glpK* gene, which encodes an enzyme of the glycerol metabolism, was found to control antibiotic tolerance and colony phase variation. This reversible phase variation is based on two successive FS mutations: first a +1 FS in *glpK* conferring tolerance and second a −1 FS restoring the original open reading frame of *glpK*. Reversible gene silencing through frameshifting is not restricted to the *glpK* gene. For example, another reversible drug

resistance mechanism mediated by FS in the *orn* gene has recently been reported in Mtb[59]. Moreover, Gupta and Alland identified 74 events in the genome of Mtb clinical isolates designated as "frame-shift scars": two FS mutations in the same gene that disrupt and subsequently restore the integrity of the gene[43]. These events have been found in 48 genes of Mtb and multiple scars were detected in the ESX-1 gene cluster encoding a secretion system important for Mtb virulence. Frequent frameshifting in homo-oligonucleotide runs of the ESX-1 gene cluster of Mtb clinical isolates has also been reported[60]. High FS incidence has also been found in PE-PPE genes[43,60] encoding secreted proteins[61]. The Mtb genome contains around 170,000 runs of three homo-oligonucleotides or more[62]. We believe that our study indicates that TLS polymerases are major contributors to the mycobacterial genome plasticity and advance DinB1 and DnaE2 as the prime mediators of homo-oligonucleotide FS mutagenesis.

In this study, we also discovered that *dinB1* and *dnaE2* have a redundant role in resistance to alkylation damage in *M. smegmatis*, which could explain the lack of phenotypes obtained with Mtb *dinB* mutants[38]. The ability of DinB to confer alkylation damage tolerance has been reported in *E. coli*[8] and for *Pseudomonas aeruginosa* and *Pseudomonas putida*, taxa in which a DnaE2 homolog also plays a role[63]. Alkylation damage can be generated by exogenous and endogenous sources[64,65]. Most endogenous sources of alkylation damage in bacteria are produced by metabolic enzymes catalyzing nitrosylation[66,67]. Mtb is exposed to nitrosative stress during macrophage infection[25,28]. A recent study reported that the pathogenic bacterium *Brucella abortus* encounters alkylating stress during macrophage infection and that the alkylation-specific repair systems are required for long-term mouse infection[68]. In Mtb, the deletion of similar alkylation-specific repair systems causes sensitivity to alkylating agents but does not impact virulence[69,70], suggesting a possible functional redundancy between alkylation-specific repair systems and the TLS polymerases. Future studies will be conducted to investigate the redundancy between DnaE2, DinBs, and the alkylation-specific repair systems in the tolerance to alkylation damage and Mtb survival during infection.

In summary, we have discovered a role of mycobacterial TLS polymerases, in particular DinB1, in alkylation damage tolerance, genome plasticity, and antibiotic resistance. By exposing the capability of DinB1 and DnaE2 to incorporate FS in homo-oligonucleotide runs, our work reveals potential molecular actors of reversible gene silencing, a recently discovered mechanism linked to antibiotic tolerance and virulence in Mtb[43,44,59].

# Methods

## Bacterial strains
Strains used in this work are listed in Supplementary Table 1. *Escherichia coli* strains were cultivated at 37 °C in Luria-Bertani (LB) medium. *M. smegmatis* strains were grown at 37 °C in Middlebrook 7H9 medium (Difco) supplemented with 0.5% glycerol, 0.5% dextrose, 0.1% Tween 80. Antibiotics were used at the following concentrations: 5 μg/ml streptomycin (Sm), 50 μg/ml hygromycin (Hyg).

## Plasmids
Plasmids and oligonucleotides used in this study are listed in Supplementary Tables 2 and 3. Plasmids were constructed in *E. coli* DH5α. For the construct of complementation plasmids, ORFs together with their 5′ flanking regions (~500 bp) were amplified by PCR using *M. smegmatis* mc²155 or *M. tuberculosis* Erdman genomic DNA as template. The PCR products were cloned into pDB60 digested with *EcoR1* using recombination-based cloning (In-Fusion, Takara). For the constructs of *dinB1* expression plasmids, ORFs were amplified using *M. smegmatis* mc²155 or *M. tuberculosis* Erdman genomic DNA and were cloned into pMSG419 digested with *ClaI*. For the *leuD* inactivation plasmid construct, the ~500 bp regions flanking the deleted nucleotides were

amplified using *M. smegmatis* mc²155 genomic DNA as template and were cloned into pAJF067 digested with *Nde1* using recombination-based cloning (In-Fusion, Takara). For plasmids carrying the *kan* gene inactivated by run of homo-oligonucleotides, the *kan* gene was amplified by PCR using the pAJF266 vector as template. The amplified fragments were cloned into pDP60 digested with *EcoRI* using recombination-based cloning (In-Fusion). The absence of mutations in constructs was verified by DNA sequencing.

## Growth and cell viability
Cells in exponential growth phase cultured without inducer were back diluted in fresh 7H9 medium supplemented with 50 nM of inducer (Anhydrotetracycline: ATc) to $OD_{600} = 0.001$. Growth was measured by monitoring $OD_{600}$ for 48 h. When $OD_{600}$ reached a value around 1, cultures were back diluted in fresh +ATc 7H9 medium to $OD_{600} = 0.001$ and measured values were multiplied by the dilution factor. For cell viability, +ATc 7H9 liquid cultures were collected by centrifugation, resuspended in −ATc 7H9 and serial dilutions were spotted and cultured on −ATc Difco Middlebrook 7H10 agar medium and incubated 48 h at 37 °C. The number of CFU was counted and the result expressed in number of CFU per Optical Density Unit (ODU: CFU in 1 ml of a culture at $OD_{600} = 1$). For growth on agar medium, cells were grown in log phase without inducer and 5 μl of serial dilutions were spotter on 7H10 medium supplemented with 0, 5, or 50 nM of ATc and incubated at 37 °C for 72 h.

## Construct of *M. smegmatis* strains
Plasmids were introduced into *M. smegmatis* by electrotransformation. The construct of unmarked deletion mutants used in this study is detailed in Dupuy et al.[53]. Unmarked 1 bp and 2 bp deletions upstream of the start codon of *leuD* were incorporated in each strain using a double recombination reaction as described in Barkan et al.[71] and using plasmids listed in Supplementary Table 2. Plasmids carrying *kan* genes inactivated by an homo-oligonucleotide run, listed in Supplementary Table 2, were introduced at the *attB* site of the *M. smegmatis* genome.

## Disc diffusion assay
Bacteria were grown to exponential phase, diluted in 3 ml of pre-warmed top agar (7H9, 6 mg/ml agar) to an $OD_{600}$ of 0.01 and plated on 7H10. A filter disc was put on the dried top agar and was spotted with 2.5 μl of 10 M $H_2O_2$, 10 mg/ml cip, 500 μg/ml mitomycin C, 100% MMS, 100 mg/ml MNNG, 50 mg/ml 4-NQO, or 100 mg/ml nitrofurazone. The diameter of the growth inhibition zone was measured after incubation for 48 h at 37 °C.

## Agar-based assay
Bacterial cultures grown to exponential phase were diluted to an $OD_{600}$ of 0.1. Serial dilutions were performed from $10^0$ to $10^{-5}$ in 7H9 and 5 μl of each dilution was plated on 7H10 or 7H10 supplemented with 0.05% MMS. Pictures were taken after 3 d incubation at 37 °C.

## UV sensitivity assay
Bacterial cultures in exponential growth phase were diluted to an $OD_{600}$ of 0.1 and serial dilutions (5 μl) were spotted on 7H10 plates. The plates were exposed to UV radiation (wavelength = 254 nm) at doses of 0, 5, 10, 15, or 20 J m⁻² using a Stratalinker 2400 UV Crosslinker (Stratagene). Plates were imaged after 3 d incubation at 37 °C.

## Immunblotting
Cell lysates were prepared from 2 ml aliquots of a log-phase culture ($OD_{600}$ of 0.4) withdrawn at 0 h, 4 h, or 24 h after ATc addition to the cultures. Cells were collected by centrifugation, resuspended in PBS buffer supplemented with 0.1% Tween 20 (PBST), lysed by incubation with 10 mg/ml lysozyme for 15 min at 37 °C, and treated with 100 mM dithiothreitol for 10 min at 95 °C. Proteins were separated by

electrophoresis in a NuPAGE™ 4–12%, Bis-Tris gel (Novex) and transferred to a PVDF membrane. Blots were blocked and probed in 5% Omniblot milk in PBST. Proteins on blots were detected using anti-RpoB (Biolegend, Cat#663905, RRID: AB_2566583) or anti-RecA (Pocono Rabbit Farm & Laboratory) antibodies incubated for 1 h at 1:10,000 dilutions and secondary Horseradish peroxidase-antibodies (#62-6520 and #65-6120, Invitrogen) at 1:1000 dilution. Blots were imaged in iBright FL1000 (Invitrogen) after treatment of the membrane with Amersham ECL western blotting detection reagents (GE Healthcare) according to manufacturer's instructions.

## Mutation frequency determination

Bacteria were grown to exponential phase in 7H9 medium from a single colony. In experiments with deletion mutants, cultures were back-diluted at an $OD_{600}$ of 0.0005 in fresh medium and cultured for 24 h. For *dinB1* expression experiments, cultures were back diluted at $OD_{600}$ of 0.004 in fresh medium supplemented with 50 nM ATc and cultured in presence of the inducer for 16 h. Cells ($OD_{600}$ ~0.5) were concentrated 20-fold by centrifugation and pellet resuspension and 100 μl of a $10^{-6}$ dilution was plated on 7H10 agar whereas 200 μl was plated on 7H10 with 100 μg/ml rif or 4 μg/ml Kan for the measurement of substitution mutations or FS mutations in homo-oligonucleotide runs, respectively. For the measurement of FS mutations in *leuD*, cells were cultivated in 7H9 medium supplemented with 50 μg/ml leucine and plated on 7H10 (200 μl) or 7H10 supplemented with 50 μg/ml leucine. For stress-induced mutagenesis, cells were treated with UV (10 J m$^{-2}$) or $H_2O_2$ (2.5 mM) for 2 h, washed and incubated for 4 h at 37 °C in fresh medium. MMS (0.010%) was added to the cultures 4 h before plating. Mutation frequency was expressed by the mean number of selected colonies per $10^8$ CFU from independent cultures. For each strain and condition, the number of independent cultures used to measure the mutation frequency is indicated by the number of gray dots in each bar of graphs. For the determination of the mutation spectrum, the RRDR of the *rpoB* gene in rif$^R$ colonies, the *kan* gene in kan$^R$ colonies or the *leuD* gene in leu$^+$ colonies, was amplified and sequenced using primers listed in Supplementary Table 3. For each strain, sequenced colonies were picked among at least six biological replicates. Mutation spectra were expressed as relative frequency, percent of mutations types found in each strain or condition, or absolute frequency, number of mutation types per $10^8$ CFU obtained by multiplying the relative frequency of the mutation by the rif$^R$, leu$^+$ or kan$^R$ frequency.

## In vitro DNA slippage assay

Recombinant *M. smegmatis* DinB1 was produced in *E. coli* and purified as described previously[40]. Protein concentration was determined by using the Bio-Rad dye reagent with bovine serum albumin as the standard. A 5′ $^{32}$P-labeled primer DNA strand was prepared by reaction of a 13-mer oligonucleotide, 5′-dCGTGTCGCCCTTC, with T4 polynucleotide kinase and [γ$^{32}$P]ATP. The labeled DNA was separated from free ATP by electrophoresis through a nondenaturing 18% polyacrylamide gel and then eluted from an excised gel slice. The primer templates for assays of DNA polymerase were formed by annealing the 5′ $^{32}$P-labeled 13-mer pDNA strand (SG-FS1) to a series of unlabeled template strands (SG-FS2-7) at 1:3 molar ratio to form the primer-templates depicted in Fig. 5c and Supplementary Fig. 6. Polymerase reaction mixtures (10 μl) containing 50 mM Tris·HCl, pH 7.5, 5 mM $MnCl_2$, 0.125 mM dNTP or ddNTP as specified, 1 pmol (0.1 μM) $^{32}$P-labeled primer-template DNA, and 10 pmol (1 μM) DinB1 were incubated at 37 °C for 15 min. The reactions were quenched by adding 10 μl of 90% formamide, 50 mM EDTA, 0.01% bromophenol blue-xylene cyanol. The samples were heated at 95 °C for 5 min and then analyzed by electrophoresis through a 40-cm 18% polyacrylamide gel containing 7.5 M urea in 44.5 mM Tris-borate, pH 8.3, 1 mM EDTA. The products were visualized by autoradiography. Where specified, the gel

was scanned with a Typhoon FLA7000 imager and the relative distributions of individual extension products were quantified with ImageQuant software.

## RT-qPCR and RNA sequencing

RT-qPCR experiments were conducted exactly as described in Adefisayo et al.[32] whereas RNAseq results were obtained from the RNAseq raw data published in Adefisayo et al.[32]. When overexpressed, *dinB1*$^{Msm}$ and *dinB1*$^{Mtb}$ expressions were measured 6 h after ATc addition.

## Quantification and statistical analysis

One-way analysis of variance (ANOVA) and a Bonferroni post-test were performed using Prism 9 software (GraphPad) on ln-transformed data for all statistical analyses of this work except for growth and viability experiments for which a 2-ways ANOVA was used.

## Reporting summary

Further information on research design is available in the Nature Research Reporting Summary linked to this article.

## Data availability

All data generated in this study are presented in the Figures and Tables. Source data are provided with this paper. Additional information required to reanalyze the data reported in this paper are available from the corresponding author upon reasonable request. Source data are provided with this paper.

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

## Acknowledgements

This work is supported by NIH (NIH grant #AIO64693) and this research was funded in part through the NIH/NCI Cancer Center Support Grant P30CA008748. P.D. was supported in part by a « Jeune Scientifique » salary award from the French National Institute of Agronomic Science (INRA). We thank all Glickman and Shuman lab members for helpful discussions. We thank Jamie Bean for reanalyzing the RNAseq data.

## Author contributions

P.D., S.G., O.A., S.S., and M.S.G. designed research; P.D., S.G., O.A., and J.B. performed research; P.D., S.G., O.A., S.S., and M.S.G. analyzed data; P.D. and M.S.G. wrote the paper, with input from S.S.

## Competing interests

M.S.G. has received consulting fees from Vedanta Biosciences, PRL NYC, and Fimbrion Therapeutics and has equity in Vedanta biosciences. All other authors declare no competing interests.
