## [Peer Review File · Nature Communications]

Distinctive roles of translesion polymerases DinB1 and DnaE2 in diversification of the mycobacterial genome through substitution and frameshift mutagenesisEditorial Note: Parts of this Peer Review File have been redacted as indicated to remove third-party material where no permission to publish could be obtained. Parts of this Peer Review File have been redacted as indicated to maintain the confidentiality of unpublished data.

REVIEWER COMMENTS

Reviewer #1 (Remarks to the Author):

Mycobacterium tuberculosis has found numerous ways to adapt to differing environments including potentially harsh conditions within their human hosts and the pressure of anti-tuberculosis drugs. Although past research has focused on the role of transcriptional modulations and single nucleotide polymorphisms in these types of adaptations, it is becoming increasingly certain that small frame shifting (FS) insertions and deletions play an important role in both transient and permanent adaptations by way of gene modulation. The work by Dupus et al is highly important in this context as it delves deeply into one fundamental mechanism that appears to govern this mutational process in mycobacteria.

The manuscript presents novel, clear and convincing data on the role of DinB1 (and to a lesser extent DnaE2) in generating small frame shift mutations. Once published, I expect that this work will be referenced along with Bishoff et al's Cell paper in future work in the field. That is a compliment. However, I do have few comments that could potentially further strengthen this otherwise admirable manuscript.

Major comments

Page 5, entire page. This section begins the investigation into the function of DinB. The "test" for function used here is simply growth attenuation in the mutants compared to the wild type control. While I agree that this phenotype might be a direct result of DinB functioning, one could come up with many other reasons why DinB overexpression could cause this same phenotype in ways unrelated to its function as a DNA polymerase. Perhaps DinB interacts non-specifically with essential proteins in addition to DNA and unrelated to replication, maybe it acts as a dominant negative on some regulatory gene, the possibilities are endless. What's more, this section is entirely unnecessary for the rest of the manuscript where DinB functionality becomes well established mechanistically both in vitro and in vivo. My advice would be to eliminate virtually the entire page, i.e. lines 74 – 102 and make minor changes to the rest of the work using the much more definitive data on mutational changes to reach all of the same conclusions.

There are a few more points that are made on page 5 that if made elsewhere in a new version are also needing additional support. For example, I don't think that the conclusion that DinB related growth defects were not caused by activation of DNA damage responses is sufficiently supported just by finding this defect still occurs in a delta *recA* background. Perhaps overexpression of DinB activates a DNA damage response in a *recA* independent manner. An RNA-sequence experiment would be an additional way to validate this point. Or one could just omit this conclusion.

The data on (quite extreme) differences in mutation selectivity in the *rpoB* region is potentially fascinating. Still, one wonders. Is there any chemical or biophysical mechanisms that might explain this or has anything like this ever been observed in the bacterial mutagenic literature? Even if not, just because its new observation doesn't mean that its not true. However, I wonder whether an artifact related to selection of siblings from one stock culture might be a more pedestrian explanation. Would the authors consider repeating this experiment again, starting from highly diluted (approx. 10K CFU) cultures (to ensure genetic uniformity) of a single colony? This type of reproducibility would be highly confirmatory.

Minor comments:

Since a lot of the phenotype relates to *dinB* overexpression (OE), the study would be more complete with something in the introduction or results about if/what induces *dinB* expression in *Mycobacterium tuberculosis*. I don't think that this would need any additional wet lab experiments. An analysis of existing RNA-Seq databases should have plenty of material for this.

Lines 156 to 159. This initially seemed like the authors were saying the same thing twice. I was eventually able to figure this out (they are not repeating themselves). But perhaps this could be stated more clearly?

Lines 195 – 196 and elsewhere. “the Δ dinB123 and Δ dnaE2 Δ dinB123 mutants” is easy to misread (as I initially did) as three separate mutants (with dinB123 twice), perhaps it would be written here and later as “the Δ dinB123 and Δ dnaE2 + Δ dinB123 double mutants”. It’s a minor point, but for a dense manuscript each point of clarity helps.

Line 204. Maybe I missed this, but I don’t recall DinB experiments being performed with streptomycin at this point. Perhaps this sentence should refer only to rifampin but not to “or streptomycin”?

Page 12 and beyond. Why were ddNTPs used here as opposed to dNTPs? I think the reason is that the ddNTPs terminate oligo elongation and give a more clear result, although this could also have been done by just reducing the CCC or GGG terminal repeat to a single nucleotide? In any case for one not familiar with the role of ddNTPs versus dNTPs, a short sentence could clarify here.

Line 291 “on the contrary” it’s not clear to me how what follows is really a contrary to the paragraph above it. In fact, this section could be more generally reworked for clarity. It’s the only place where I really had to puzzle over what the authors were trying to say.

Not sure why the section starting at line 257 is only performed with dinB and dnaE2 OE strains while the section starting with line 315 is only performed with dinB and dnaE2 knockout (KO) strains? Wouldn’t you normally perform the experiments described in both sections with both OE and KO strains and look for opposite effects? I certainly understand that this strategy does not always work out so nicely, especially with the KOs, when there are redundancies among different genes, but if this is the case I still would want the opposite (and less successful) experiment to be briefly described with these perhaps less than clear results – at least in the supplement. As is we are left a bit puzzled why the complete set of experiments were not done.

Line 331. It’s interesting that that the dinB123 KO should show such strong sequence specificity. Would the authors perhaps speculate how generating FSs with 4Cs but not 4Gs might have a biological impact through strand selectivity?

Reviewer #2 (Remarks to the Author):

The ms by Dupuy et al. describes mainly the role of DinB1 from *Mycobacterium smegmatis* and *M. tuberculosis*. It also looks at other DinBs and compares the results to DnaE2, the currently known translesion DNA polymerase in *Mycobacterium*. The authors show that DinB1 is also an error prone DNA polymerase in both *M. smegmatis* and *M. tuberculosis*. They discover that the *M. tuberculosis* DinB1 previously tested had a deletion of 5 Aa at the N-terminal end. There is a comprehensive study of mutagenesis by using a Rif^S-Rif^R assay and determining then the mutational signature of these DNA pols as well as frameshifts (both -1 and +1). This is a thorough analysis of these DNA polymerases with data supporting the authors’ statements.

The main gap in the ms is that all the in vivo analysis, including mutagenesis, is carried out with plasmids of unknown copy number and level of expression when induced by ATc. The authors do mention in the ms that *M. tuberculosis* DinB1 is highly expressed when in the host, which would suggest that the results shown here are relevant. However, there is no data to support this assumption. In addition, most of the findings are consistent with previous findings about DinB-like DNA polymerases, which the authors only sometimes acknowledge (see below for details).

Detailed comments

Lines 16-19. In the abstract the two sentences describing the activities of DinB1 and DnaE2 and other DinBs are confusing. It is not clear from the abstract what each of these DNA pols do. I suggest making it into a single sentence with clear differences.

Line 36. DinB is not as mutagenic as the authors make it to believe here. UmuC is the most mutagenic DNA pol in *E. coli*. Authors should tone down language (see Jarosz et al.).

Line 61. when you say "active" you mean they function as DNA polymerases? TLS DNA pols are distributive, not very active. Precise language will make this clearer.

Line 71. This is an example of a good sentence to include in the abstract. The activities of the comparing DNA pols are clearly stated.

Line 74. I understand why authors decided to highlight the finding of extra Aa needed for DinB1 activity. However, I disagree. It is not really an "extension" but rather an error in the orf of Mtb. The Mss DinB also has the 5 Aa, so it may be something unique to Mycobacterium DinBs. I suggest highlighting that Mycobacterium DinBs have extra Aa and moreover these had not been previously considered for the Mtb DinB. Any ideas? probably folding related. You could use alpha fold to see the differences between *E. coli* DinB (with a shorter N-terminal end) and Mycobacterium DinB1s. In the Suppl. Fig 1e the D8- corresponding Asp in Mtb and Mss should also be highlighted as "metal binding" since this residue is part of DinB's active site, made up of Aa at the beginning of the ORF in *E. coli*, e.g., D8, needed for formation of the phosphodiester bond. Finally, the expression of Mtb is under an ATc regulated promoter. What's the in vivo relevance if this is overproduced? Is there a sense for how much protein is being made in WT Mtb compared to the Atc derived one?

Line 102. This is the same phenotype found in *E. coli*. It should be mentioned here. It is still unclear why this is the case, but it is not necessarily because of access to the beta clamp. The DinB1 with a deletion of the beta clamp binding motif may be an unstable protein, easily degraded in vivo. The authors have not tested that the DinB1 Δ B-clamp derivative is as stable as DinB1. I suggest tampering the language.

Line 105. The authors here and in other parts of the ms refer to DinB1 "intrinsic flexibility" needed for lesion bypass. The overall enzyme may be intrinsically flexible though that fact has not been shown. What is known is that the active site of TLS DNA pols must be flexible, and it is known that they are larger providing thus the ability to fit in the active site a lesion on the template. Replicative DNA pols have instead very tight active sites that permit to check for the correct geometry in base pairing. It is a very clever and simple way to check that the correct base pair has been introduced. I suggest adding that the flexibility is in the active site. There are multiple references to this aspect in the literature.

Line 112-113. Again, the issue about the beta clamp binding motif. I suggest softening the statement. Add a "suggest" to the sentence.

Line 117. The conclusion may be a stretch. The authors have shown that DinB1 is only mutagenic at high copy number. The only way to reconcile this is to measure how much dinB1 gene is expressed in the conditions tested and compare it to the overexpression in the host. If this is not possible, then temper the language.

Line 126. It would be good here to present the data in the context of what mutations have been found clinically for Mtb. Suppl. Table 6 has the info.

Line 133. I suggest moderating the language as the authors do not really know how mutagenic DinB1 is in vivo as your system uses DinB1 overproduction.

Lines 152-157. The authors here present the data about DNA damage induced mutagenesis. There is a problem with the UV-induced mutagenesis since it increased by 100-fold in the WT strain though there is a decrease of only 10x in the dnaE2 deletion mutant. It is likely that there is another factor playing a role here. So, in the conclusion in line 167 I agree with the H₂O₂ statement, but not with the UV-Induced one.

Line 170. Another mention to the "flexibility" that should be amended.

Line 182-183. The word "severe" is extreme. The observed difference is ~20% worse. Tone it down.

Lines 189-190. There does not seem to be full complementation. There is partial complementation in all cases, i.e., there is no WT phenotype.

Lines 196-197. As above there does not seem to be full complementation. The figure title should be changed as NFZ is not a form of alkylation damage.

Line 233. This observation is known for E. coli DinB. A reference should be added here or noted that this is not news but expected.

Line 291. The experiments were not done with a mix of dNTPs as suggested by the text, but in the presence of the upcoming complementary nucleotide. There is no data shown for all dNTPs. The DinB1 in the run of As plus G is not as efficient, but it is pretty good with the Ts and Cs. Precise language helps.

Line 312 and above. There is mention of percentages of unextended template. How were this percentages calculated?

Section starting in line 315. Most of these findings are described previously in the ms. The data here are a comparison with the other DinBs and DnaE2. I suggest having this as a single section. The same conclusions can be reached. The ms is getting to be overly long for no reason.

Line 338. Conclusion 3...or in stressed cells. the experiments are with dinB overproducers. Need to measure DinB cellular concentration to make that assertion.

Line 386. Authors mention that dinB1 expression is induced in pulmonary TB. This key to make the assertions throughout the ms. How close is dinB1 expression in the system used by the authors?

Line 406. The statement here is well known for other DinB-like enzymes. Authors should reference this here. There are many references for E. coli DinB addressing this fact.

Reviewer #3 (Remarks to the Author):

Summary:

Mycobacterium tuberculosis, the causative agent of tuberculosis acquires drug resistance-conferring mutations through mutagenesis of drug targets encoded in the bacterial chromosome. As a result, spontaneous and stress induced mechanisms of mutagenesis, together with processes that maintain genome stability, have been the subject of prior study. Particularly interesting have been members of the DinB/Y-family of DNA polymerases that mediate chromosomal mutagenesis through translesion synthesis in other well-studied model organisms. In contrast, the role of mycobacterial DinB

polymerases in mutagenesis and emergence of drug resistance has been unclear. Rather, DnaE2, an alternative DNA polymerase has been implicated in mutagenesis. In this work, Dupuy and colleagues explore the role of DinB1 in translesion synthesis and find that over-expression of DinB1 from *M. smegmatis* resulted in cell death, together induction of RecA and an increase in rifampin resistance. Expression of the *M. tuberculosis* homologue, with an additional 5 amino acids added on, compared to previous annotations, also resulted in a similar effect. Overexpression of a catalytically dead version of DinB1 exacerbated the growth defect whilst a derivative of DinB1 lacking the B-clamp binding motif did not cause any growth defects or cell death. These data suggest that DinB1 competed for the replisome with the replicative polymerase. The growth and mutagenesis effects were maintained in a *recA* or a *dnaE2* deletion mutant, confirming that these effects were independent of a generalized DNA-damage response and also not associated with previously established role of DnaE2 in mutagenesis. Sequencing of the RRDR in rifampin resistant mutants indicated that increased expression of DinB1 was associated with A>G or T>C transition mutations, however, deletion of *dnaE2* and/or all three DinB homologues did not alter the prevalence of spontaneous mutation types when compared to the wild type strain. DNA-damage induced mutagenesis using different mutagens revealed a role for DnaE2 in mediated certain types of mutations but no specific role was identified for DinB homologues in these experiments. Loss of all four translesion polymerases resulted in increased sensitivity to MNNG (resulting in alkylation damage to DNA), a defect that was reversed upon genetic complementation with *dnaE2* or *dinB1* (from *M. smegmatis* or *M. tuberculosis*). To investigate the role of translesion polymerases in tolerance to N2-dG adducts, the authors exposed their mutants to 4-NQO and NFZ. They found that the Δ *dinB123* and Δ *dnaE2* Δ *dinB123* mutants were highly sensitive to NFZ, this defect was reversed upon genetic complementation with *dnaE2*, *DinB1* (from *M. smegmatis* or *M. tuberculosis*) or *DinB3*. Using a frameshift reporter assay that was dependent on reversion to leucine prototrophy, the authors demonstrated that increased expression of *DinB1* resulted in elevated -1 frameshift mutations in homo-oligonucleotide runs. Similar results were obtained using a kanamycin resistance reporter system with homo-polymeric runs and using an in vitro primer extension assay. In vivo, -1 frameshifts, as measured by the leucine auxotrophy assay were reduced in the Δ *dinB1*, Δ *dinB123* and Δ *dnaE2* Δ *dinB123* mutants, this was reversed by complementation with *dinB1*. For reasons that are unclear, this effect did not manifest with the kanamycin reporter system. Finally, the authors demonstrate that DnaE2 plays the major role in -2 frameshift mutations in response to DNA-damage. This is a carefully conducted study, with well-constructed arguments. There are some concerns that should be addressed:

1. A substantive growth and survival defect was noted upon overexpression of *dinB* genes in *M. smegmatis* but no expression data are provided (plus minus tet) to indicate the order by which expression was elevated in the tet system used. Providing expression data for a perturbed system is standard in the field and should be done.
2. Similar to point 2, expression analysis for complementation experiments in Figure 4 must be provided for all *dinB* homologues that were tested. To confirm the conclusion that certain *dinB* genes could not reverse defects, expression data should be provided to indicate that these were expressed at levels comparable to homologues that were able to reverse defects.
3. The in vitro experiments should contain a catalytic deficient mutant. This is standard to ensure that no translesion polymerase co-purified, even at low amounts, with the recombinant protein.
4. Why do the complementation assays in Figure 5B only include *dinB1*? Did the authors test if the other homologues did not work? This is important.
5. A point that is not addressed is the induction of RecA upon over-expression of *dinB* genes. This did not point to a generalized DNA damage response and the observation is not sufficiently resolved. Perhaps it is better to remove this, but keep the data from the *recA* knockout?

Minor

Figure 1a and b, and elsewhere. X-axis in minutes seems somewhat strange. This would reflect better as hours. Not sure what 3000 minutes means.

The Legend for Figure 4 does not describe what the panels are depicting.

Reviewer #1:

Major comment 1:

Page 5, entire page. This section begins the investigation into the function of DinB. The “test” for function used here is simply growth attenuation in the mutants compared to the wild type control. While I agree that this phenotype might be a direct result of DinB functioning, one could come up with many other reasons why DinB overexpression could cause this same phenotype in ways unrelated to its function as a DNA polymerase. Perhaps DinB interacts non-specifically with essential proteins in addition to DNA and unrelated to replication, maybe it acts as a dominant negative on some regulatory gene, the possibilities are endless. What’s more, this section is entirely unnecessary for the rest of the manuscript where DinB functionality becomes well established mechanistically both in vitro and in vivo. My advice would be to eliminate virtually the entire page, i.e. lines 74 – 102 and make minor changes to the rest of the work using the much more definitive data on mutational changes to reach all of the same conclusions.

- We thank reviewer #1 for these comments. We agree that growth defect measurement is not the central phenotype of DinB1 or the most relevant to our conclusions. However, a deleterious effect of cell growth is one of the most dramatic *dinB* OE phenotype reported in *E. coli* (Uchida et al., 2008), which was not reproducible in the previous *Mtb dinB* study (Kana et al., 2010). In this context, these experiments introduce the discovery of an active form of *Mtb* DinB1 using this growth phenotype. On page 5 we removed “and measured bacterial growth” from the sentence: “To investigate the role of DinB1 in mycobacteria, we expressed *M. smegmatis dinB1* (*dinB1*^{Msm}) from an Anhydrotetracycline (ATc) inducible promoter (tet promoter) and measured bacterial growth.”.
- We also agree with reviewer #1 that we did not prove that the growth defect is due to a direct effect on replication, although the reversal of this phenotype by mutation of the B clamp binding motif does suggest this conclusion. On pages 5, we modified the conclusion: “These results suggest that DinB1 interacts with the replicative machinery in vivo and competes with the replicative DNA polymerase at replication forks, as proposed in *E. coli* (Furukohri et al., 2008; Uchida et al., 2008). We cannot exclude the possibility that DinB1^{ΔB-clamp} derivative is not as stable as WT and that the DinB1-dependent growth defect is unrelated to replication.”.
- The two first paragraphs were combined in one and the title was changed for “DinB1^{Mtb} activity requires five N-terminal amino acids omitted from the annotated ORF.”.

Major comment 2:

There are a few more points that are made on page 5 that if made elsewhere in a new version are also needing additional support. For example, I don’t think that the conclusion that DinB related growth defects were not caused by activation of DNA damage responses is sufficiently supported just by finding this defect still occurs in a delta *recA* background. Perhaps overexpression of DinB activates a DNA damage response in a *recA* independent manner. An RNA-sequence experiment would be an additional way to validate this point. Or one could just omit this conclusion.

- We agree with reviewer #1 that we cannot exclude the possibility of a RecA-independent DNA damage response. On page 5, we removed the “However, the effect of DinB1 on growth was not due to activation of the DNA damage response as it was preserved in the *ΔrecA* background (Supplementary Fig. 1c).” sentence and the associated figure.
- On page 6, we added “RecA-dependent DNA damage response” to the sentence: “We observed a similar induction of mutagenesis after *dinB1*^{Msm} expression in *ΔrecA* and *ΔdnaE2* backgrounds (Supplementary Fig. 1d), showing that the effect of *dinB1* on mutation frequency is not the

consequence of the **RecA-dependent DNA damage response** or the previously defined role of DnaE2 in mutagenesis(Boshoff et al., 2003), further strengthening the conclusion that DinB1 **can be** directly mutagenic.”.

Major comment 3:

The data on (quite extreme) differences in mutation selectivity in the *rpoB* region is potentially fascinating. Still, one wonders. Is there any chemical or biophysical mechanisms that might explain this or has anything like this ever been observed in the bacterial mutagenic literature? Even if not, just because its new observation doesn't mean that its not true. However, I wonder whether an artifact related to selection of siblings from one stock culture might be a more pedestrian explanation. Would the authors consider repeating this experiment again, starting from highly diluted (approx. 10K CFU) cultures (to ensure genetic uniformity) of a single colony? This type of reproducibility would be highly confirmatory.

- We thank the reviewer for this comment and we were similarly concerned with such an artifactual result when designing this experiment. We designed the experiment to reduce the risk of this bias:
 - o Each independent biological replicate (symbolized by gray dots in fig. 2a) was started from an independent single colony and not a common stock culture.
 - o The mutation profile we report was established by sequencing *rif^R* colonies from several biological replicates, each derived from a single colony and not from a single stock culture. On page 24 we added the sentence: “**For each strain, sequenced colonies were picked among at least six biological replicates.**”.
- To further respond to this concern and demonstrate that the mutation profile observed when *dinB1* is expressed is due to the activity of the polymerase and not an inoculum bias, we repeated this experiment with and without the inducer for 15h. If the enhancement of A>G or C>T mutations is due to an inoculum effect unrelated to DinB1 activity, the mutation spectrum should change in the no ATC condition. This control was not presented in the original figure. The experiment has been performed four times and, for each condition, 4 *rif^R* per biological replicate were sequenced except for the tet-*dinB1^{Msm}* +ATC condition for which 3 *rif^R* per biological replicate were sequenced. Inducer addition did not change the *rif^R* frequency or the *rpoB* mutation spectrum in bacteria carrying the empty vector. Whereas, in absence of inducer, bacteria carrying *dinB1*-inducible vectors had similar *rif^R* frequency than the control strain, inducer addition increased the *rif^R* frequency as well as the proportion of the A>G or C>A mutations detected among *rif^R* in bacteria carrying *dinB1*-inducible vectors. For each strain, colors of dots show similar inoculum. We are presenting this figure here for the reviewers but have not incorporated it into the SI figures.

[redacted]

Minor comments:

Since a lot of the phenotype relates to *dinB* overexpression (OE), the study would be more complete with something in the introduction or results about if/what induces *dinB* expression in *Mycobacterium tuberculosis*. I don't think that this would need any additional wet lab experiments. An analysis of existing RNA-Seq databases should have plenty of material for this.

- On page 4, we added the sentence “Whereas *Mtb dinB2* expression is enhanced by novobiocin (Boshoff et al., 2004), *Mtb dinB1* is part of the SigH regulon (Kaushal et al., 2002) and its expression is induced by rifampicin (Boshoff et al., 2004) and during human pulmonary TB infection (Rachman et al., 2006).”.

Lines 156 to 159. This initially seemed like the authors were saying the same thing twice. I was eventually able to figure this out (they are not repeating themselves). But perhaps this could be stated more clearly?

- On page 7, we modified the sentence “UV- and H₂O₂-induced mutagenesis was DnaE2-dependent, declining by 10- and 4-fold in $\Delta dnaE2$ cells compared to WT, whereas MMS-induced mutagenesis was not impacted (Fig. 3a).” by “Induction of the *rif^R* frequency by UV and H₂O₂ was DnaE2-dependent, declining by 10- and 4-fold in $\Delta dnaE2$ cells compared to WT, whereas MMS-induced mutagenesis was not impacted (Fig. 3a).”
- The mutation types (G>A or C>T, A>C or T>G and G>C or C>G) were added to the sentence “All types of UV- and H₂O₂-induced *rif^R* mutations were reduced by the *dnaE2* deletion (Fig. 3b)”.

Lines 195 – 196 and elsewhere. “the $\Delta dinB123$ and $\Delta dnaE2 \Delta dinB123$ mutants” is easy to misread (as I initially did) as three separate mutants (with *dinB123* twice), perhaps it would be written here and later as “the $\Delta dinB123$ and $\Delta dnaE2 + \Delta dinB123$ double mutants”. It's a minor point, but for a dense manuscript each point of clarity helps.

- We changed “ $\Delta dnaE2 \Delta dinB123$ ” for “ $\Delta dnaE2 dinB123$ ” to avoid confusion.

Line 204. Maybe I missed this, but I don't recall *DinB* experiments being performed with streptomycin at this point. Perhaps this sentence should refer only to rifampin but not to “or streptomycin”?

- corrected

Page 12 and beyond. Why were ddNTPs used here as opposed to dNTPs? I think the reason is that the ddNTPs terminate oligo elongation and give a more clear result, although this could also have been done by just reducing the CCC or GGG terminal repeat to a single nucleotide? In any case for one not familiar with the role of ddNTPs versus dNTPs, a short sentence could clarify here.

- The reviewer is correct. As suggested, on page 12, we have added a clarifying sentence as follows: “ddNTPs are employed to force termination upon incorporation of the first templated nucleotide following the homo-oligomeric tract.”

Line 291 “on the contrary” it's not clear to me how what follows is really a contrary to the paragraph above it. In fact, this section could be more generally reworked for clarity. It's the only place where I really had to puzzle over what the authors were trying to say.

- On page 12, We deleted the confusing phrase “on the contrary.” We also clarified the quantification of the unextended products in the ddCTP-containing reactions in Fig. 5e, as follows: “In the case of

the T tract templates, we see that the fraction of products that fail to be extended in the presence of ddCTP, defined as $\frac{\text{unextended product}}{\text{unextended product} + \text{extended product}}$, increases progressively as the template tract lengthens from 4T (13% unextended) to 6T (34%) to 8T (48%) (Fig. 5e). This result suggests a contribution of -1 slippage (rather than pure failure to incorporate ddC), the reasoning being that lengthening the template homo-oligonucleotide run is expected to enhance slippage but not impact DinB1's ability to incorporate ddNTPs."

Not sure why the section starting at line 257 is only performed with *dinB* and *dnaE2* OE strains while the section starting with line 315 is only performed with *dinB* and *dnaE2* knockout (KO) strains? Wouldn't you normally perform the experiments described in both sections with both OE and KO strains and look for opposite effects? I certainly understand that this strategy does not always work out so nicely, especially with the KOs, when there are redundancies among different genes, but if this is the case I still would want the opposite (and less successful) experiment to be briefly described with these perhaps less than clear results – at least in the supplement. As is we are left a bit puzzled why the complete set of experiments were not done.

- The section starting at line 257 (1st submitted version) describes in vitro studies, but we suspect that the review is referring to the section starting at line 202 of the 1st submitted version entitled "DinB1 mediates -1 frameshift mutations." *dinB1* and *dnaE2* deletion mutants had several phenotypes (alkylation sensitivity, substitution mutagenesis and FS mutagenesis) whereas *dinB2* and *dinB3* deletions did not. The OE experiments were performed with *dinB1* to corroborate these results but OE of *dnaE2* had no effect (data not shown).

Line 331. It's interesting that that the *dinB123* KO should show such strong sequence specificity. Would the authors perhaps speculate how generating FSs with 4Cs but not 4Gs might have a biological impact through strand selectivity?

- We agree with the reviewer that the difference observed between 4C and 4G is intriguing. We performed preliminary experiments to test a putative strand specificity for the DinB1 activity which were non-conclusive. We prefer not to develop this point in this current paper.

Reviewer #2:

General comment + three detailed comments:

The main gap in the ms is that all the in vivo analysis, including mutagenesis, is carried out with plasmids of unknown copy number and level of expression when induced by ATc. The authors do mention in the ms that *M. tuberculosis* DinB1 is highly expressed when in the host, which would suggest that the results shown here are relevant. However, there is no data to support this assumption.

Line 74. the expression of Mtb is under an ATc regulated promoter. What's the in vivo relevance if this is overproduced? Is there a sense for how much protein is being made in WT Mtb compared to the Atc derived one?

Line 117. The conclusion may be a stretch. The authors have shown that DinB1 is only mutagenic at high copy number. The only way to reconcile this is to measure how much *dinB1* gene is expressed in the conditions tested and compare it to the overexpression in the host. If this is not possible, then temper the language.

Line 133. I suggest moderating the language as the authors do not really know how mutagenic DinB1 is in vivo as your system uses DinB1 overproduction.

Line 386. Authors mention that *dinB1* expression is induced in pulmonary TB. This key to make the assertions throughout the ms. How close is *dinB1* expression in the system used by the authors?

- We thank reviewer #2 for his comments. We agree that some of important results of the manuscript were obtained by overexpressing *dinB1*, particularly the ability of DinB1 to confer rifampicin resistance which is not detectable using deletion mutants (fig. 2 and fig 3.). Although deduced from overexpression, the results highlight the in vivo mutagenic potential of DinB1 and reveal a distinct mutation signature from DnaE2 and these mutations are found in clinical Mtb strains, suggesting that they are relevant to physiologic DinB1 expression. The two other important functions of DinB1 we reveal (tolerance to alkylation damage and ability to incorporate -1 frameshift mutations) were demonstrated by both gain and loss of function (deletion mutants (Fig. 4, Fig. 6, and Fig. 7), together with complementary overexpression and in vitro experiments for FS mutagenesis (Fig. 5)), indicating that many of our conclusions are relevant to physiologic DinB1 function.
- We agree with reviewer#2 that the level of induction after ATc treatment was missing. We performed RTqPCR to show that *dinB1* expression is strongly induced by inducer treatment. These data have been added (Supplementary Fig. 1a) and on page 5 we added “**Addition of ATc enhanced the level of *dinB1*^{Msm} mRNA by 1000-fold (Supplementary Fig. 1a).**” was added and the sentence “**Expression of the mRNA encoding this longer form of the Mtb DinB1 (*dinB1*^{Mtb+5aa}) was induced 250-fold by ATc addition (Supplementary Fig. 1a) and impaired *M. smegmatis* growth (Fig. 1d), suggesting that the first five amino acids of DinB1 are essential for in vivo activity.**”.
- We agree with reviewer#2 that the level of induction of *dinB1* in host should be given and compared to the level of induction of our experiments. The paragraph “**Our study shows that DinB1 confers rifampicin resistance when the gene is highly expressed. Transcriptomic studies reported a 4.5-fold induction of *dinB1*^{Mtb} caused by rifampicin treatment of Mtb (Boshoff et al., 2004) and a 12-fold induction of the gene in pulmonary TB compared to culture conditions (Rachman et al., 2006). Further experiments will be needed to test if DinB1 participates to rifampicin resistance acquisition during infection.**” was added on page 16 (line 386 of the 1st submission manuscript).

- We incorporated several modifications to temper the conclusion about the role of DinB1 in substitution mutagenesis (rifampicin resistance acquisition)
 - o On page 6 (line 117 of the 1st submitted version) the conclusion was modified: “We observed a similar induction of mutagenesis after *dinB1*^{Msm} expression in $\Delta recA$ and $\Delta dnaE2$ backgrounds (Supplementary Fig. 1d), showing that the effect of *dinB1* on mutation frequency is not the consequence of the **RecA-dependent DNA damage response** or the previously defined role of DnaE2 in mutagenesis (Boshoff et al., 2003), further strengthening the conclusion that DinB1 **can be** directly mutagenic.”.
 - o On page 7 (line 133 of the 1st submitted version) the sentence “The intrinsic mutagenicity of DinB1 demonstrated above supports a role for the enzyme in chromosomal mutagenesis in the absence of exogenous DNA damage.” was modified for “The intrinsic mutagenicity of DinB1 demonstrated above supports a role for the enzyme in chromosomal mutagenesis **at high copy number.**”

Detailed comments:

Lines 16-19. In the abstract the two sentences describing the activities of DinB1 and DnaE2 and other DinBs are confusing. It is not clear from the abstract what each of these DNA pols do. I suggest making it into a single sentence with clear differences.

- The abstract was modified as suggested.

Line 36. DinB is not as mutagenic as the authors make it to believe here. UmuC is the most mutagenic DNA pol in *E. coli*. Authors should tone down language (see Jarosz et al.).

- On page 3, The word “**highly**” was removed from the sentence: “In *E. coli*, DinB and UmuDC are **highly** mutagenic, inducing substitution mutations as well as indels (Kato and Nakano, 1981; Kim et al., 1997, 2001; Napolitano et al., 2000; Steinborn, 1978; Wagner and Nohmi, 2000).”

Line 61. when you say "active" you mean they function as DNA polymerases? TLS DNA pols are distributive, not very active. Precise language will make this clearer.

- On page 4, the sentence was modified for: “*M. smegmatis* DinB1, DinB2, and DinB3 **catalyze DNA-templated primer extension in vitro** (Ordenez and Shuman, 2014; Ordenez et al., 2014).”.

Line 74. I understand why authors decided to highlight the finding of extra Aa needed for DinB1 activity. However, I disagree. It is not really an "extension" but rather an error in the orf of Mtb. The Mss DinB also has the 5 Aa, so it may be something unique to Mycobacterium DinBs. I suggest highlighting that Mycobacterium DinBs have extra Aa and moreover these had not been previously considered for the Mtb DinB. Any ideas? probably folding related. You could use alpha fold to see the differences between *E. coli* DinB (with a shorter N-terminal end) and Mycobacterium DinB1s. In the Suppl. Fig 1e the D8- corresponding Asp in Mtb and Mss should also be highlighted as “metal binding” since this residue is part of DinB’s active site, made up of Aa at the beginning of the ORF in *E. coli*, e.g., D8, needed for formation of the phosphodiester bond.

- On page 5, the title of the first paragraph was modified for: “**DinB1^{Mtb} activity requires five N-terminal amino acids omitted from the annotated ORF.**”.
- The alignment of DinBs sequences presented in Supplementary Fig. 2 (Supplementary Figure 1 in the 1st submitted version) does not really show a conserved N’ter extension for mycobacterial DinBs compared to the *E. coli* protein. However, it shows a conservation of an Arginine residue in *E. coli*,

M. smegmatis and *M. tuberculosis*, absent in the original annotation of the Mtb gene. This residue alone could be essential for the activity or the stability of the DNA polymerase.

- The D8 residue was highlighted as “metal binding”.

Line 102. This is the same phenotype found in *E. coli*. It should be mentioned here. It is still unclear why this is the case, but it is not necessarily because of access to the beta clamp. The DinB1 with a deletion of the beta clamp binding motif may be an unstable protein, easily degraded in vivo. The authors have not tested that the DinB1 Δ B-clamp derivative is as stable as DinB1. I suggest tempering the language.

Line 112-113. Again, the issue about the beta clamp binding motif. I suggest softening the statement. Add a “suggest” to the sentence.

- We agree with reviewer #2 that the absence of quantification of the protein level of DinB1 Δ B-clamp complicates our interpretation. We tried to quantify a Streptavidin tagged DinB1 inducible by ATc, together with DinB2 and DinB3 (the two other *M. smegmatis* DinB paralogues) by western blot. Unfortunately, although we detected DinB2 and DinB3, DinB1 was not visible, for reasons that we cannot determine at present. We do not have an antibody to the native protein.

- We measured by RTqPCR the level of mRNAs of *dinB1* and *dinB1* Δ B-clamp and found that both genes are induced similarly (6h of treatment), although we recognize that this experiment does not address the protein stability of the mutant.

- On page 5, the conclusion has been modified to temper the conclusion and cite *E. coli* result: “These results **suggest** that DinB1 interacts with the replicative machinery in vivo and competes with the replicative DNA polymerase at replication forks, **as proposed in *E. coli* (Furukohri et al., 2008; Uchida et al., 2008). We cannot exclude the possibility that DinB1^{ΔB-clamp} derivative is not as stable as WT and that the DinB1-dependent growth defect is unrelated to replication**”.
- On page 6, “**showing**” was replaced by “**suggesting**” in the sentence: “The expression of *dinB1*^{Msm} or *dinB1*^{Mtb+5aa} increased the frequency of rif^R by 6- or 8-fold but the expression of *dinB1*^{Δβclamp} had no effect, **showing suggesting** that an interaction between DinB1 and the replicative machinery is required for DinB1-dependent mutagenesis.”.
- On page 5, The title “DinB1 competes with the replicative polymerase for interaction with the β clamp at the replication fork.” was removed.

Line 105. The authors here and in other parts of the ms refer to DinB1 “intrinsic flexibility” needed for lesion bypass. The overall enzyme may be intrinsically flexible though that fact has not been shown. What is known is that the active site of TLS DNA pols must be flexible, and it is known that they are larger providing thus the ability to fit in the active site a lesion on the template. Replicative DNA pols have instead very tight active sites that permit to check for the correct geometry in base pairing. It is a very clever and simple way to check that the correct base pair has been introduced. I suggest adding that the flexibility is in the active site. There are multiple references to this aspect in the literature.

- “**intrinsic flexibility**” was replaced by “**active site flexibility**” on pages 3, 6, and 8.

Line 126. It would be good here to present the data in the context of what mutations have been found clinically for Mtb. Suppl. Table 6 has the info.

- This information is discussed in the discussion on page 16: “Here we report that DinB1 can induce rifampicin resistance through a mutagenic activity. In contrast to the diverse mutation spectrum of DnaE2, DinB1 confers rif^R through a unique *rpoB* mutation (CAC>CGC; H442R). This mutation has been detected in rif^R Mtb clinical isolates at a frequency between 0.8 and 5%, depending on the study (Cavusoglu et al., 2004; Hirano et al., 1999; Matsiota-Bernard et al., 1998; Qian et al., 2002; Rudeeaneksin et al., 2021; WHO mutations catalogue, 2021; Williams et al., 1998) (Supplementary Tables 4, 6).”. We understand the proposition of reviewer#2 but we prefer limit redundancy in the text and not talk about clinical data here.

Lines 152-157. The authors here present the data about DNA damage induced mutagenesis. There is a problem with the UV-induced mutagenesis since it increased by 100-fold in the WT strain though there is a decrease of only 10x in the dnaE2 deletion mutant. It is likely that there is another factor playing a role here. So, in the conclusion in line 167 I agree with the H2O2 statement, but not with the UV-Induced one.

- On page 8, the sentence was changed for: “These results show that DnaE2, but not DinBs, **mediates contributes to** UV- and H2O2-induced substitution mutagenesis with a distinct mutation spectrum from the DinB1 signature.”

Line 170. Another mention to the “flexibility” that should be amended.

- Corrected.

Line 182-183. The word “severe” is extreme. The observed difference is ~20% worse. Tone it down.

- On page 8, “severe” was replaced by “higher”.

Lines 189-190. There does not seem to be full complementation. There is partial complementation in all cases, i.e., there is no WT phenotype.

- On page 8, “partially” was added to the sentence: “The $\Delta dnaE2 \Delta dinB123$ sensitivity to MMS and MNNG was ~~also~~ partially complemented by an ectopic copy of $dinB1^{Mtb}$ but not $dinB2^{Mtb}$ (Fig. 4b, d), indicating a conservation of DinB1 activities between fast- and slow-growing mycobacteria.”.

Lines 196-197. As above there does not seem to be full complementation. The figure title should be changed as NFZ is not a form of alkylation damage.

- On page 9, “partially” was added to the sentence: “Ectopic expression of $dinB1^{Msm}$, $dinB3^{Msm}$, $dnaE2^{Msm}$ or $dinB1^{Mtb}$ in the $\Delta dnaE2 \Delta dinB123$ partially reversed the NFZ sensitivity (Fig. 4f), reinforcing the substantial redundancy of translesion polymerases in mycobacteria for bypassing damage”.
- The title of Fig. 4 and the title of the paragraph on page 8 was corrected as suggested.

Line 233. This observation is known for *E. coli* DinB. A reference should be added here or noted that this is not news but expected.

- On page 8, *E. coli* references were added to the sentence: “We next investigated the role of DnaE2 and DinBs in the tolerance to alkylation damage, as reported for *E. coli* DinB (Bjedov et al., 2007; Jarosz et al., 2006), by testing the sensitivity of *M. smegmatis* TLS polymerase mutants to the chemical methylating agents MMS and methylnitrosoguanidine (MNNG)”

Line 291. The experiments were not done with a mix of dNTS as suggested by the text, but in the presence of the upcoming complementary nucleotide. There is no data shown for all dNTPs. The DinB1 in the run of As plus G is not as efficient, but it is pretty good with the Ts and Cs. Precise language helps.

- The section on in vitro assays of DinB1 does not include reference to a “mix” of dNTPs. Rather, we refer to “the presence of various nucleotide substrates and combinations thereof” when introducing the data shown in Fig. S6 (page 12). We believe our phrasing is precise regarding the experiments in Figs. 5e and S6, whereby we make clear that we are providing the next complementary nucleotide after the homo-oligomeric tract in the reactions containing a single dNTP plus a ddNTP. There is no need to show data of “all dNTPs” because the templates employed contain only two of the four nucleobases. We show in Fig. S6 that DinB1 catalyzed six steps of dTMP addition to the A6 template in the presence of dTTP and that inclusion of dGTP elicited three further steps of dGMP addition opposite the 5'-terminal CCC element of the template strand. This results shows that DinB1 is competent for fill-in synthesis.

Line 312 and above. There is mention of percentages of unextended template. How were this percentages calculated?

- On page 13, the calculation method was added to the sentence: “In the case of the T tract templates, we see that the fraction of products that fail to be extended in the presence of ddCTP, defined as $\frac{\text{unextended product}}{\text{unextended product plus extended product}}$, increases progressively as the template tract lengthens from 4T (13% unextended) to 6T (34%) to 8T (48%).”

Section starting in line 315. Most of these findings are described previously in the ms. The data here are a comparison with the other DinBs and DnaE2. I suggest having this as a single section. The same conclusions can be reached. The ms is getting to be overly long for no reason.

- We respectfully disagree with reviewer#2, this section demonstrates the role of DinB1 in FS mutagenesis under physiological conditions (we compared WT to TLSpol deletion mutants), whereas the previous sections showed the mutagenicity of DinB1 when expressed at high copy number. We believe that this entire section responds to the major comment of reviewer#2 discussed above about the use of gain and loss of function models.

Line 338. Conclusion 3...or in stressed cells. the experiments are with *dinB* overproducers. Need to measure DinB cellular concentration to make that assertion.

- Results of this section were obtained with WT and *dinBs* mutants, not *dinBs* overproducers.

Line 406. The statement here is well known for other DinB-like enzymes. Authors should reference this here. There are many references for *E. coli* DinB addressing this fact.

- On page 17 we added “**Because eukaryal translesion polymerase fidelity can be impacted by the PCNA clamp (Maga et al., 2007)** and because we found here that DinB1 mutagenesis depends on its β clamp interaction, we speculate that slippage might be increased if the β clamp was tethering DinB1 to the template.”

Reviewer #3:

1. A substantive growth and survival defect was noted upon overexpression of *dinB* genes in *M. smegmatis* but no expression data are provided (plus minus tet) to indicate the order by which expression was elevated in the tet system used. Providing expression data for a perturbed system is standard in the field and should be done.

- We thank reviewer #3 for these comments. This comment is the same than Reviewer#2 1st comment. **Please see the response to reviewer#2.**

2. Similar to point 2, expression analysis for complementation experiments in Figure 4 must be provided for all *dinB* homologues that were tested. To confirm the conclusion that certain *dinB* genes could not reverse defects, expression data should be provided to indicate that these were expressed at levels comparable to homologues that were able to reverse defects.

- Results presented in Figure 4 are complementations of the quadruple *dnaE2dinB123* mutant using an integrative vector carrying a *dinB* homologue or *dnaE2* expressed under their native promoters. We do not expect that each homologue is expressed at similar levels, they are regulated according to their native promoter. The following sentence was added to the fig. 4 legend: **“Translesion polymerases were expressed under their native promoter from a genome integrated vector in panels b, d and f.”**

3. The in vitro experiments should contain a catalytic deficient mutant. This is standard to ensure that no translesion polymerase co-purified, even at low amounts, with the recombinant protein.

- We showed previously in Ordonez et al., 2014 that a DinB1 active site mutant D113A has no primer extension activity, i.e., the preps are not contaminated with a co-purifying *E. coli* polymerase.

[redacted]

4. Why do the complementation assays in Figure 5B only include *dinB1*? Did the authors test if the other homologues did not work? This is important.

- Experiment presented in Figure 5B are not complementation assays but overexpression of *dinB1* in a WT background.
- We did similar experiments with *dinB2* and *dinB3* which will be presented in a future publication.

5. A point that is not addressed is the induction of RecA upon over-expression of *dinB* genes. This did not point to a generalized DNA damage response and the observation is not sufficiently resolved. Perhaps it is better to remove this, but keep the data from the *recA* knockout?

- The sentence on page 5 was modified: “*dinB1*^{Msm} expression also **caused DNA damage**, as evinced by an increase in the steady-state level of the RecA protein at 4 h post-induction by ATc (Fig. 1c).”
- As also pointed by reviewer#1, we cannot exclude the possibility of a RecA-independent DNA damage response.
- On page 5, we removed the “**However, the effect of DinB1 on growth was not due to activation of the DNA damage response as it was preserved in the $\Delta recA$ background (Supplementary Fig. 1c).**” sentence and the associated figure (Fig. 1d).
- On page 6, we added “**RecA-dependent DNA damage response**” to the sentence: “We observed a similar induction of mutagenesis after *dinB1*^{Msm} expression in $\Delta recA$ and $\Delta dnaE2$ backgrounds (Supplementary Fig. 1d), showing that the effect of *dinB1* on mutation frequency is not the consequence of the **RecA-dependent DNA damage response** or the previously defined role of DnaE2 in mutagenesis(Boshoff et al., 2003), further strengthening the conclusion that DinB1 **can be** directly mutagenic.”.

Minor

Figure 1a and b, and elsewhere. X-axis in minutes seems somewhat strange. This would reflect better as hours. Not sure what 3000 minutes means.

- Corrected

The Legend for Figure 4 does not describe what the panels are depicting

- Corrected

References:

- Bjedov, I., Dasgupta, C.N., Slade, D., Le Blastier, S., Selva, M., and Matic, I. (2007). Involvement of *Escherichia coli* DNA polymerase IV in tolerance of cytotoxic alkylating DNA lesions in vivo. *Genetics* 176, 1431–1440. <https://doi.org/10.1534/genetics.107.072405>.
- Boshoff, H.I.M., Reed, M.B., Barry, C.E., and Mizrahi, V. (2003). DnaE2 polymerase contributes to in vivo survival and the emergence of drug resistance in *Mycobacterium tuberculosis*. *Cell* 113, 183–193. [https://doi.org/10.1016/s0092-8674\(03\)00270-8](https://doi.org/10.1016/s0092-8674(03)00270-8).
- Boshoff, H.I.M., Myers, T.G., Copp, B.R., McNeil, M.R., Wilson, M.A., and Barry, C.E. (2004). The transcriptional responses of *Mycobacterium tuberculosis* to inhibitors of metabolism: novel insights into drug mechanisms of action. *J Biol Chem* 279, 40174–40184. <https://doi.org/10.1074/jbc.M406796200>.
- Cavusoglu, C., Karaca-Derici, Y., and Bilgic, A. (2004). In-vitro activity of rifabutin against rifampicin-resistant *Mycobacterium tuberculosis* isolates with known *rpoB* mutations. *Clin Microbiol Infect* 10, 662–665. <https://doi.org/10.1111/j.1469-0691.2004.00917.x>.
- Furukohri, A., Goodman, M.F., and Maki, H. (2008). A dynamic polymerase exchange with *Escherichia coli* DNA polymerase IV replacing DNA polymerase III on the sliding clamp. *J Biol Chem* 283, 11260–11269. <https://doi.org/10.1074/jbc.M709689200>.
- Hirano, K., Abe, C., and Takahashi, M. (1999). Mutations in the *rpoB* Gene of Rifampin-Resistant *Mycobacterium tuberculosis* Strains Isolated Mostly in Asian Countries and Their Rapid Detection by Line Probe Assay. *J Clin Microbiol* 37, 2663–2666. .
- Jarosz, D.F., Godoy, V.G., Delaney, J.C., Essigmann, J.M., and Walker, G.C. (2006). A single amino acid governs enhanced activity of DinB DNA polymerases on damaged templates. *Nature* 439, 225–228. <https://doi.org/10.1038/nature04318>.
- Kana, B.D., Abrahams, G.L., Sung, N., Warner, D.F., Gordhan, B.G., Machowski, E.E., Tsenova, L., Sacchettini, J.C., Stoker, N.G., Kaplan, G., et al. (2010). Role of the DinB Homologs Rv1537 and Rv3056 in *Mycobacterium tuberculosis*. *Journal of Bacteriology* 192, 2220–2227. <https://doi.org/10.1128/JB.01135-09>.
- Kato, T., and Nakano, E. (1981). Effects of the *umuC36* mutation on ultraviolet-radiation-induced base-change and frameshift mutations in *Escherichia coli*. *Mutat Res* 83, 307–319. [https://doi.org/10.1016/0027-5107\(81\)90014-2](https://doi.org/10.1016/0027-5107(81)90014-2).
- Kaushal, D., Schroeder, B.G., Tyagi, S., Yoshimatsu, T., Scott, C., Ko, C., Carpenter, L., Mehrotra, J., Manabe, Y.C., Fleischmann, R.D., et al. (2002). Reduced immunopathology and mortality despite tissue persistence in a *Mycobacterium tuberculosis* mutant lacking alternative sigma factor, SigH. *Proc Natl Acad Sci U S A* 99, 8330–8335. <https://doi.org/10.1073/pnas.102055799>.
- Kim, S.-R., Maenhaut-Michel, G., Yamada, M., Yamamoto, Y., Matsui, K., Sofuni, T., Nohmi, T., and Ohmori, H. (1997). Multiple pathways for SOS-induced mutagenesis in *Escherichia coli*: An overexpression of *dinB/dinP* results in strongly enhancing mutagenesis in the absence of any exogenous treatment to damage DNA. *Proc Natl Acad Sci U S A* 94, 13792–13797. .
- Kim, S.R., Matsui, K., Yamada, M., Gruz, P., and Nohmi, T. (2001). Roles of chromosomal and episomal *dinB* genes encoding DNA pol IV in targeted and untargeted mutagenesis in *Escherichia coli*. *Mol Genet Genomics* 266, 207–215. <https://doi.org/10.1007/s004380100541>.
- Maga, G., Villani, G., Crespan, E., Wimmer, U., Ferrari, E., Bertocci, B., and Hübscher, U. (2007). 8-oxo-guanine bypass by human DNA polymerases in the presence of auxiliary proteins. *Nature* 447, 606–608. <https://doi.org/10.1038/nature05843>.

Matsiota-Bernard, P., Vrioni, G., and Marinis, E. (1998). Characterization of *rpoB* Mutations in Rifampin-Resistant Clinical *Mycobacterium tuberculosis* Isolates from Greece. *Journal of Clinical Microbiology* <https://doi.org/10.1128/JCM.36.1.20-23.1998>.

Napolitano, R., Janel-Bintz, R., Wagner, J., and Fuchs, R.P.P. (2000). All three SOS-inducible DNA polymerases (Pol II, Pol IV and Pol V) are involved in induced mutagenesis. *EMBO J* *19*, 6259–6265. <https://doi.org/10.1093/emboj/19.22.6259>.

Ordonez, H., and Shuman, S. (2014). *Mycobacterium smegmatis* DinB2 misincorporates deoxyribonucleotides and ribonucleotides during templated synthesis and lesion bypass. *Nucleic Acids Res.* *42*, 12722–12734. <https://doi.org/10.1093/nar/gku1027>.

Ordonez, H., Uson, M.L., and Shuman, S. (2014). Characterization of three mycobacterial DinB (DNA polymerase IV) paralogs highlights DinB2 as naturally adept at ribonucleotide incorporation. *Nucleic Acids Res* *42*, 11056–11070. <https://doi.org/10.1093/nar/gku752>.

Qian, L., Abe, C., Lin, T.-P., Yu, M.-C., Cho, S.-N., Wang, S., and Douglas, J.T. (2002). *rpoB* genotypes of *Mycobacterium tuberculosis* Beijing family isolates from East Asian countries. *J Clin Microbiol* *40*, 1091–1094. <https://doi.org/10.1128/JCM.40.3.1091-1094.2002>.

Rachman, H., Strong, M., Ulrichs, T., Grode, L., Schuchhardt, J., Mollenkopf, H., Kosmiadi, G.A., Eisenberg, D., and Kaufmann, S.H.E. (2006). Unique transcriptome signature of *Mycobacterium tuberculosis* in pulmonary tuberculosis. *Infect Immun* *74*, 1233–1242. <https://doi.org/10.1128/IAI.74.2.1233-1242.2006>.

Rudeeaneksin, J., Phetsuksiri, B., Nakajima, C., Bunchoo, S., Suthum, K., Tipkrua, N., Fukushima, Y., and Suzuki, Y. (2021). Drug-resistant *Mycobacterium tuberculosis* and its genotypes isolated from an outbreak in western Thailand. *Trans R Soc Trop Med Hyg* *115*, 886–895. <https://doi.org/10.1093/trstmh/traa148>.

Steinborn, G. (1978). Uvm mutants of *Escherichia coli* K12 deficient in UV mutagenesis. I. Isolation of uvm mutants and their phenotypical characterization in DNA repair and mutagenesis. *Mol Gen Genet* *165*, 87–93. <https://doi.org/10.1007/BF00270380>.

Uchida, K., Furukohri, A., Shinozaki, Y., Mori, T., Ogawara, D., Kanaya, S., Nohmi, T., Maki, H., and Akiyama, M. (2008). Overproduction of *Escherichia coli* DNA polymerase DinB (Pol IV) inhibits replication fork progression and is lethal. *Mol Microbiol* *70*, 608–622. <https://doi.org/10.1111/j.1365-2958.2008.06423.x>.

Wagner, J., and Nohmi, T. (2000). *Escherichia coli* DNA polymerase IV mutator activity: genetic requirements and mutational specificity. *J Bacteriol* *182*, 4587–4595. <https://doi.org/10.1128/JB.182.16.4587-4595.2000>.

WHO mutations catalogue (2021). Catalogue of mutations in *Mycobacterium tuberculosis* complex and their association with drug resistance.

Williams, D.L., Spring, L., Collins, L., Miller, L.P., Heifets, L.B., Gangadharam, P.R.J., and Gillis, T.P. (1998). Contribution of *rpoB* Mutations to Development of Rifamycin Cross-Resistance in *Mycobacterium tuberculosis*. *Antimicrob Agents Chemother* *42*, 1853–1857. .

REVIEWERS' COMMENTS

Reviewer #1 (Remarks to the Author):

I have reviewed the revised manuscript and the authors responses to my initial comments. I believe that the authors have adequately addressed all of my concerns, and I do not see any new concerns in the revision.

Reviewer #2 (Remarks to the Author):

The revised ms has answered most of the questions that reviewers posed to the authors. Whenever unable to answer directly to a question, they did address the question finding a solution or taming and changing the language in the ms.

The ms addresses an important unanswered question in MTb mutagenesis, in which resistance to antibiotic arises exclusively from chromosomal mutations.

Reviewer #3 (Remarks to the Author):

The authors have adequately addressed my concerns.